# Speech-in-noise discriminability after noise exposure: Insights from a gerbil model of acoustic trauma

Carolin Jüchter[1,2], Rainer Beutelmann[1,2], Sonny Bovee[1,2], Katja Bleckmann[1,2], Georg Martin Klump[1,2]*

**1** Division Animal Physiology and Behavior, Department of Neuroscience, School of Medicine and Health Sciences, Carl von Ossietzky University Oldenburg, Lower Saxony, Oldenburg, Germany, **2** Cluster of Excellence "Hearing4all"

* georg.klump@uni-oldenburg.de

## Abstract

Speech comprehension, especially in the presence of background sounds, allegedly declines as a consequence of noise-induced hearing loss. However, the connection between noise overexposure and deteriorated speech-in-noise perception despite normal audiometric thresholds (hidden hearing loss) is not yet clear. This study investigates speech-in-noise discrimination in young-adult Mongolian gerbils before and after an acoustic trauma to examine the link between noise exposure and speech-in-noise perception. Nine young-adult gerbils were trained to discriminate a deviant consonant-vowel-consonant combination (CVC) or vowel-consonant-vowel combination (VCV) in a sequence of CVC or VCV standards, respectively. The logatomes were spoken by different speakers and masked by a steady-state speech-shaped noise. After the gerbils obtained the behavioral baseline data, they underwent an acoustic trauma and participated in the behavioral experiments again. Applying multidimensional scaling, response latencies were used to generate perceptual maps reflecting the gerbils' internal representations of the sounds pre- and post-trauma. To evaluate how the discrimination of vowels and consonants was altered after noise exposure, changes in response latencies between phoneme pairs were investigated in relation to their articulatory features. Numbers of intact inner hair cell synapses were counted, and auditory brainstem responses were measured to assess peripheral auditory function. Perceptual maps of vowels and consonants were very similar before and after noise exposure. Interestingly, the gerbils' overall vowel discrimination ability was improved after the acoustic trauma, even though the gerbils suffered from noise-induced hearing loss with a temporary threshold shift for frequencies above 4 kHz. In contrast, there were only minor changes in the gerbils' consonant discrimination ability. Moreover, noise exposure showed a differential influence on response latencies for vowel and consonant discriminations depending on the articulatory features. Altogether, the results show that an acoustic trauma followed by a temporary

**Data availability statement:** The underlying data is freely accessible via zenodo (https://doi.org/10.5281/zenodo.15489868).

**Funding:** This work was funded by the Deutsche Forschungsgemeinschaft (DFG, German Research Foundation) under Germany's Excellence Strategy – EXC 2177/1 (Project-ID 390895286). The funders had no role in study design, data collection and analysis, decision to publish, or preparation of the manuscript.

**Competing interests:** The authors have declared that no competing interests exist.

threshold shift is not necessarily linked to speech-in-noise perception difficulties associated with hidden hearing loss.

## Introduction

About 19.3% of the worldwide population is affected by hearing loss [1]. One of the main causes for hearing loss is the exposure to loud sounds, with approximately 16% of the cases in adults being attributed to occupational noise [2]. Additionally, the World Health Organization estimates that more than 50% of young people between the ages of 12 and 35 put themselves at risk of hearing damage by listening to music at too high volumes on their personal audio devices [3]. The resulting noise-induced hearing loss (NIHL) was hypothesized to be causal for difficulties in speech perception, especially in the presence of background sounds. When such deficits in speech-in-noise perception are not reflected in an altered pure tone audiometry, the condition is referred to as hidden hearing loss [4]. However, the connection between noise overexposure and a deteriorated speech-in-noise perception is not clear yet and potential underlying mechanisms are still under debate.

While some studies reported an association between noise exposure and deteriorated speech-in-noise perception in combination with normal hearing thresholds in human listeners [4,5], other studies did not find such a connection [e.g., 6–9] (for review, see [10]). An often-suggested potential physiological cause of noise-induced speech processing difficulties is synaptopathy – the loss or pathological change of synapses between inner hair cells (IHCs) and auditory nerve fibers (ANFs). However, a clear link between noise exposure and synaptopathy in human listeners could not be demonstrated. While it is very difficult to assess synaptopathy in humans, and results from previous studies are conflicting [10], animal studies provided evidence that noise exposure can indeed cause synaptopathy in combination with normal audiometric thresholds [11], potentially due to a selective loss of low and medium spontaneous rate (SR) ANFs [12,13]. This loss of low-SR ANFs was suggested to be reflected in a reduction of the amplitude of wave I of the auditory brainstem response (ABR) [12,14–16]. However, in contrast to these observations from rodent studies, contradicting results were found in humans, with some studies reporting reduced ABR wave I amplitudes in listeners with a history of noise overexposure [17,18], while others did not find such a connection [e.g., 9,19,20]. Thus, neither the link between noise exposure and synaptopathy nor between noise exposure and speech processing difficulties is verified in human listeners. Because of these variations and the limited opportunities for research on NIHL in humans, the use of an appropriate animal model is needed. In contrast to the purely correlation-based investigations on effects from barely characterizable noise traumata on speech perception in human listeners, animal studies offer the possibility for a precise experimental approach with a well-defined noise exposure. Further, animals can serve as their own controls in a pre- vs. post-trauma comparison, offering better interpretability in comparison to the methodologies available in human studies.

In rodents, there is evidence for a connection between noise exposure and synaptopathy, however, it is not known how synaptopathy affects speech-in-noise processing. In order to fill this gap of knowledge, we investigated in the present study how noise exposure affects the behavioral speech sound discrimination ability of Mongolian gerbils (*Meriones unguiculatus*). Gerbils have been commonly used as an animal model for NIHL [e.g., 13,21–25] and for research on speech sound processing [e.g., 26–31]. Perceptual maps and discrimination patterns for vowels and consonants have been found to be similar in young gerbils and young human listeners [31,32]. Because of these similarities, gerbils are suggested to be a promising model organism for investigating the effects of noise overexposure on speech-in-noise perception.

To this end, we used a behavioral paradigm in which gerbils were trained with reward-based operant conditioning to discriminate different consonant-vowel-consonant combinations (CVCs) and vowel-consonant-vowel combinations (VCVs). After the gerbils completed the behavioral baseline experiments, they underwent an acoustic trauma before they collected data in the behavioral experiments again. Thus, we investigated how the discriminability of different types of vowels and consonants changed from before to after noise overexposure. Peripheral auditory function of the gerbils was evaluated via ABRs before and after the acoustic trauma. In this way, we aim at untangling the potential connections between noise exposure, synaptopathy and speech-in-noise perception difficulties.

## Materials and methods

### Animals

A main cohort of nine young-adult Mongolian gerbils (*Meriones unguiculatus*) was used for the behavioral experiments and underwent the noise exposure and accompanying ABR-measurements. ABR data from five additional young-adult gerbils, which were part of another gerbil cohort that underwent the same acoustic trauma and ABR measurement scheme without being trained for behavioral experiments, were pooled with the ABR data from the main cohort in order to increase the animal number (and thus statistical power). All gerbils were born and raised in the animal facilities of the University of Oldenburg and originated from gerbils obtained from Charles River laboratories. The animals were housed alone or in groups of up to three gerbils of the same sex in EU type IV cages containing litter, paper towels, cardboard and paper tubes as cage enrichment. In order to increase the motivation of the gerbils participating in the behavioral experiments, they were food-deprived for the period of training and data acquisition. Consequently, they were given restricted amounts of rodent dry food outside of the experiments in addition to the custom-made 10-mg pellets that they received as rewards during the experimental sessions. Training took place five days a week and the gerbils had unlimited access to water. The gerbils' body weights were kept at about 90% of their free feeding weights and their general condition was checked on a daily basis. The gerbils were used as their own controls and participated two times in the same behavioral paradigm – once before undergoing an acoustic trauma and a second time after undergoing the acoustic trauma. During the phase of pre-trauma data collection, the gerbils were between four and nine months old. During post-trauma data collection, the gerbils were between seven and thirteen months old. Note that data for the post-trauma VCV conditions of one gerbil is missing, since it did not complete data collection. The pre-trauma data was also used as part of a comparison of behavioral data from young-adult and quiet-aged gerbils in a recently published study [33] and small parts of the behavioral datasets, that is, data for behavioral discriminations between the vowels /a:/, /e:/ and /i:/ with the phonetic context /b/ from five gerbils pre-trauma were used for a comparison with data from single-ANF recordings in a previously published study [30]. The care and treatment of the animals as well as all experimental procedures were reviewed and approved by the Niedersächsisches Landesamt für Verbraucherschutz und Lebensmittelsicherheit (LAVES), Lower Saxony, Germany, under permit number AZ 33.19-42502-04-21/3821. All procedures were performed in compliance with the NIH Guide on Methods and Welfare Consideration in Behavioral Research with Animals [34], and all efforts were made to minimize suffering.

## Auditory brainstem response

For evaluating peripheral auditory function before and after noise trauma, ABRs were measured in all gerbils ($n = 14$) at three different time points. The first ABR measurement was done directly before the gerbils underwent the acoustic trauma. The second and third ABRs were measured two days and three weeks after noise exposure to capture both temporary and permanent changes in peripheral auditory function, respectively. Initially, the gerbils were anesthetized by an intraperitoneal injection of a mixture of fentanyl (0.005% fentanyl, 0.03 mg/kg body weight), medetomidine (0.1% medetomidine, 0.15 mg/kg body weight) and midazolam (0.5% midazolam, 7.5 mg/kg body weight). Anesthesia was then maintained by subcutaneous injections of one-third dose of the initial mixture. Prior to the recordings, the gerbils received a subcutaneous injection of 2 ml saline in order to prevent dehydration, and oxygen supply (0.6 l/min) was provided during the measurements. A feedback-controlled homeothermic blanket (Harvard Apparatus) was used to maintain the animals' body temperature at approximately 37°C. The ABR measurements were performed inside of a sound-attenuating chamber (IAC 401-A, Industrial Acoustics Company). During the recordings, the head of the gerbil was fixed using a bite bar. Ear bars containing calibration microphones (ER7-C, Etymotic Research) and speakers (IE800, Sennheiser) were placed in front of the ear canals. The stainless-steel needle recording electrode was placed subcutaneously at the vertex of the skull and the reference electrode was placed subcutaneously on the midline in the neck. Both electrodes were moistened with saline solution in order to ensure low impedances. Before starting the recordings, the acoustic system was calibrated in situ by measuring the speakers' frequency characteristics while presenting a sine sweep (0.1–22 kHz, logarithmic scaling at 1 octave/s). The speakers' output during the recordings was then corrected by a minimum phase finite impulse response filter (512th order) that was derived from the impulse responses, leading to flat output levels (±3 dB) for frequencies between 0.3 and 19 kHz. ABRs were recorded in response to clicks (0.2–15 kHz, 40 µs duration) and pure tones (2, 4 and 8 kHz, 289 Hz bandwidth, 5 ms duration including 2.5 ms on/off cosine ramps) with 10-dB level steps (500 repetitions per level). All stimuli were generated using custom-written software in MATLAB (MathWorks), produced at 48 kHz sampling rate by an external audio interface (Hammerfall DSP Multiface II, RME), and preamplified (HB7, Tucker Davis Technologies) before presentation. Finally, ABRs were amplified (10,000 times) and bandpass filtered (0.3–3 kHz) by an amplifier (ISO 80, World Precision Instruments), and digitized using the external audio card (48 kHz sampling rate). After the ABR measurement, the gerbil was either kept anesthetized for noise exposure (see section Acoustic trauma) or the anesthesia was antagonized with a mixture of naloxone (0.04% naloxone, 0.5 mg/kg body weight), atipamezole (0.5% atipamezole, 0.375 mg/kg body weight) and flumazenil (0.01% flumazenil, 0.4 mg/kg body weight). ABR thresholds were defined in 5 dB steps as the lowest sound level at which clear ABR waves were still visually distinguishable. All ABR measures reported here are based on the mean threshold, amplitude or latency of both ears of each animal.

## Acoustic trauma

All gerbils ($n = 14$) were exposed to a noise trauma in order to provoke NIHL. The gerbils were between four and ten months old at the time of the noise exposure. For the gerbils that participated in the behavioral experiments, this corresponds to the time when they had completed all conditions of the behavioral experiments for the first time. Directly before the acoustical traumatization, an ABR measurement was carried out, enabling an evaluation of the pre-trauma status of the peripheral auditory system. After the ABR measurement and before starting the noise exposure, the anesthesia of the gerbil (see section Auditory brainstem response) was maintained through a subcutaneous injection of one-third to two-third dose of the initial mixture of anesthesia (depending on the time since the last injection). The animal was then placed in a custom-made box with an acoustically optimized sound field that was designed to avoid standing waves. During noise exposure, the gerbil's body temperature was maintained at approximately 37°C using a feedback-controlled homeothermic blanket (Harvard Apparatus) and two cameras allowed for visual control of the animal. The noise exposure comprised 2 hours of stimulation with a 2–4 kHz octave band noise at 115 dB SPL. Stimulation was done using four loudspeakers

(Plus XS.2, Canton) driven by a power amplifier (STA-1508, IMG Stage Line) and connected via a signal processor (RX6, Tucker-Davis Technologies) to a Linux computer (OptiPlex 5040, Dell) with custom written software. The noise exposure setup was calibrated with a sound level meter (2238 Mediator, Brüel & Kjær) prior to each traumatization by measuring – and if necessary, adjusting – the noise level. After the noise exposure, the gerbils' anesthesia was antagonized with a mixture of naloxone (0.04% naloxone, 0.5 mg/kg body weight), atipamezole (0.5% atipamezole, 0.375 mg/kg body weight) and flumazenil (0.01% flumazenil, 0.4 mg/kg body weight).

## Setup for behavioral experiments

The setup for the behavioral experiments was the same as used in previous studies (for details, see [31]). In brief, experiments took place in three functionally equivalent setups that were situated in sound-attenuating chambers with reverberation times ($T_{30}$) between 12 and 30 ms. A custom-built elongated platform with a pedestal in the middle was positioned in the center of each setup, approximately one meter above the ground. At the front end of the platform, there was a food bowl connected to an automatic feeder, facing a loudspeaker that was used for acoustic stimulation. Light barriers detected the movements of the gerbil on the platform and its position on the pedestal, and an infrared camera above the platform allowed for additional visual control of the animal during the experiments, which were performed in darkness. The behavioral setup was calibrated with a sound level meter two times a week for a reference frequency of 1 kHz.

## Behavioral paradigm

Nine gerbils were trained to perform the behavioral experiments employing a go/no-go paradigm that was described in detail in [31]. Operant conditioning with food pellets as positive reinforcement was used to train the gerbils to perform an oddball target detection task, in which they had to detect a deviating logatome in a sequence of reference logatomes that were repeated every 1.3 seconds. The gerbils learned to wait on the pedestal in the experimental setup to initiate a new trial. After a random waiting time between one and seven seconds, the next reference logatome was replaced by a target logatome. When the gerbil detected a target logatome, it had to jump off the pedestal within 1.5 seconds from the onset of the target stimulus in order to being rewarded with a food pellet. The training procedure took between 3 and 8 weeks for the different gerbils, starting with the discrimination of simple pure tones, slowly increasing the complexity of the stimuli that needed to be discriminated until logatomes were used. Response latencies and hit rates for the discrimination between all target and reference logatomes were recorded. Catch trials, in which the target logatome was the same as the reference logatome, were used to determine a false alarm rate as a measure of spontaneous responding. The proportion of catch trials amounted to 30.8% and 26.7% in the CVC and VCV conditions, respectively. One experimental session typically lasted between 20 and 60 minutes and each gerbil usually participated in 1–3 sessions per day.

## Stimuli

The stimulus set used in the present study was the same as in previous studies (for details, see [31]) and comprised 40 CVCs and 36 VCVs originating from the Oldenburg logatome speech corpus (OLLO) [35]. CVCs were composed of one of the medial vowels /a/, /aː/, /ɛ/, /eː/, /ɪ/, /iː/, /ɔ/, /oː/, /ʊ/ or /uː/, combined with one of the consonants /b/, /d/, /s/ or /t/ as the initial and final phoneme. VCVs consisted of one of the consonants /b/, /d/, /f/, /g/, /k/, /l/, /m/, /n/, /p/, /s/, /t/ and /v/ in the middle of the logatome, in combination with either /a/, /ɪ/ or /ʊ/ as the initial and final vowel. The initial and final phonemes within each logatome were identical (e.g., /bab/ as a CVC or /aba/ as a VCV). Only a change in the medial phoneme of the logatomes had to be detected so that the discriminability was tested between logatomes with the same phonetic context. For example, in a CVC condition with the consonant /b/ as the phonetic context and a sequence with the reference logatome /bab/, a target logatome with a different medial vowel (e.g., /bɪb/) had to be detected (/bab/→/bab/→/bɪb/→/bab/). Thus, the four CVC conditions (each with a different consonant as the phonetic context) were used to test the

discriminability of vowels, while the gerbils had to discriminate between consonants in the three VCV conditions (each with a different vowel as the phonetic context). All logatomes were used both as target and reference logatomes and the order of target and reference logatomes was randomized between animals and across sessions. The logatomes were spoken by four German speakers (two females and two males) and included two tokens per speaker. The token and speaker for each presented reference repetition and the target logatome were randomly chosen. Consequently, only a change in the medial phoneme of the logatome, not speaker identity, needed to be reported by the gerbils. Logatomes were presented at 65 dB sound pressure level (SPL) against a continuous steady-state noise masker with a speech-shaped spectrum (ICRA-1) [36] at 5 dB signal-to-noise ratio (SNR).

## Synapse counts

Established protocols were followed for histology and synapse quantification [37–39]. The gerbils were euthanized with pentobarbital-sodium (Narkodorm; 18.23% pentobarbital, 486 mg/kg body weight). In brief, following transcardial perfusion, cochleae were fixed in 4% paraformaldehyde, decalcified in 0.5 M EDTA, blocked with 3% bovine serum albumin to prevent non-specific binding, and permeabilized with 1% Triton X-100. The following primary antibodies were applied: Anti-Myosin VIIa to label hair cells (IgG polyclonal rabbit; Proteus Biosciences; cat. no. 25e6790; RRID: AB_10015251; diluted 1:400), anti-CtBP2 (C-terminal binding protein) to label presynaptic ribbons (IgG1 monoclonal mouse; BD Biosciences; cat. no. 612044; RRID: AB_399431; diluted 1:400), and anti-GluA2 to label postsynaptic receptor patches (IgG2a monoclonal mouse; Millipore; cat. no. MAB397; RRID: AB_2113875; diluted 1:400). Secondary antibodies matching the host species of the primaries were used: Goat anti-mouse (IgG1)-AF488 (Molecular Probes Inc.; cat. no. A21121; RRID: B_141514; diluted 1:1000), goat anti-mouse (IgG2a)-AF568 (Invitrogen; cat. no. A-21134; RRID: AB_10393343; diluted 1:500), and donkey anti-rabbit-AF647 (polyclonal; Life Technologies-Molecular Probes; cat. no. A-31573; RRID: AB_162544; diluted 1:1000). The cochleae were dissected and mounted on microscope slides using Vectashield Mounting Medium (H-1000, Vector Laboratories). Confocal image stacks were acquired using a Leica TCS SP8 system (Leica Microsystem CMS GmbH) and deconvolved using Huygens Essential software (Scientific Volume Imaging, version 24.04). Synapses were analyzed at five cochlear locations, corresponding to 1, 2, 4, 8, and 16 kHz [40]. From five hair cells per cochlear location, functional synapses, defined as the co-localization of pre- and postsynaptic markers, were manually counted using ImageJ [37–39,41]. In a number of cochleae, all hair cells detached during processing, possibly due to insufficient fixation, and were therefore excluded from further analysis. Thus, synapses counts could only be assessed for nine of the fourteen gerbils. If both cochleae of an animal were analyzed, synapse counts were averaged between the two ears.

## Data analysis

Details about the methods used for the analysis of the behavioral data are described in [31]. During the experiments, response latencies were measured for the discrimination between all combinations of reference and target logatomes. These response latencies were filled into confusion matrices, which were entered into the multidimensional scaling (MDS) procedure PROXSCAL [42] in SPSS (IBM, version 29). MDS was used to translate the differences in response latencies into perceptual distances in a multidimensional space representing the perceived logatome similarity by spatial proximity. In these perceptual maps, long response latencies are represented by short perceptual distances, since they correspond to a poor behavioral discriminability between two logatomes. Short response latencies are reflected by long perceptual distances indicating a good behavioral discriminability between the logatomes. The "Dispersion Accounted For" (DAF), which can range from 0 to 1 with high values indicating a better fit, was used as a goodness-of-fit measure for the perceptual maps. It can be derived from the normalized raw stress (DAF = 1 − normalized raw stress), providing a measure for the proportion of the variance in the response latencies that is explained by the distances in the MDS solution [43]. For the vowels, two-dimensional perceptual maps were used, whereas three-dimensional perceptual maps were used for

the consonants. A higher dimensionality was needed for the perceptual maps of the consonants in order to reach similar goodness-of-fit values for the perceptual maps of vowels and consonants, with more than 93% of explained variance in the MDS solutions. In addition to that, Spearman's rank correlations were calculated to compare response latencies between pre- and post-trauma data. Apart from response latencies, also hit rates and false alarm rates were recorded. The sensitivity-index *d'* was calculated for each subject and CVC or VCV condition for quantifying the subjects' discrimination ability, applying the inverse cumulative standard normal distribution function $\Phi^{-1}$ to the hit rate *(H)* and false-alarm rate *(FA)*: *d'*$= \Phi^{-1}$*(H)* $- \Phi^{-1}$*(FA)* [44]. As a validity criterion and to prevent floor effects, the average *d'* of a complete experimental session had to be larger than 0.5, otherwise the session had to be repeated.

## Statistics

Statistical analyses were carried out in SPSS (IBM, version 29). Shapiro-Wilk tests were used to test for normality of datasets. For investigating trauma-related differences in various ABR parameters, either repeated-measures ANOVAs or Friedman-tests were used, depending on the statistical distribution of the underlying dataset. For the behavioral data, two-way repeated-measures ANOVAs were used to test for differences in *d'*-values, response latencies and mean Spearman's rank correlations of response latencies between different experimental conditions and between pre- vs- post-trauma data (within-subjects factors). Two-way repeated-measures ANOVAs were also used to test for differences in synapse numbers between acoustically traumatized and non-traumatized gerbils and between different cochlear locations (within-subjects factor). Whenever the sphericity assumption for a within-subjects factor was violated (based on Mauchly's test), a Greenhouse-Geisser correction was applied to the results. Post-hoc testing was done whenever necessary via Bonferroni-corrected paired *t*-tests following ANOVAs or Dunn-Bonferroni post-hoc tests after Friedman-tests. The threshold for significance (alpha) was set to 0.05 in all statistical tests. For all boxplots, the box shows the interquartile range, which contains the middle 50% of the data, with a line indicating the median. The whiskers represent the range of data points that fall within 1.5 times the interquartile range.

## Results

### ABRs and synapse counts were negatively affected by noise exposure

ABR measurements were done in all gerbils ($n = 14$) directly before, two days after and three weeks after noise exposure in order to evaluate their baseline peripheral auditory function as well as temporary and permanent changes due to the noise exposure. Fig 1 shows the gerbils' ABR thresholds to clicks and pure tones at 2, 4 and 8 kHz, which were based on the mean threshold of both ears of each animal at the different time points (median difference between left and right ear: 5 dB). The results showed significant main effects of noise trauma and stimulus as well as an interaction effect between both (repeated-measures ANOVA, factor noise trauma: $F(2, 26) = 16.386$, $p < 0.001$, factor stimulus: $F(2.003, 26.043) = 36.173$, $p < 0.001$, noise trauma x stimulus: $F(6, 78) = 2.346$, $p = 0.039$; Fig 1). Bonferroni-corrected post-hoc testing showed that the observed interaction effect between noise trauma and stimulus was based on significant increases in mean ABR thresholds to clicks ($M_{Diff} = -7.679$, 95%-CI[−11.238, −4.119], $p < 0.001$) and 8 kHz tones ($M_{Diff} = -12.143$, 95%-CI[−18.027, −6.259], $p < 0.001$) from pre-trauma to two days post-trauma, as well as a significant decrease in mean ABR thresholds to clicks from two days post-trauma to three weeks post-trauma ($M_{Diff} = 7.321$, 95%-CI[2.523, 12.120], $p = 0.003$). Thus, the noise exposure caused a significant elevation of the ABR thresholds to clicks and 8 kHz pure tones. Three weeks post-trauma, when the remaining threshold shifts would presumably be permanent, there were only non-significant threshold shifts that amounted on average to 0.4 dB and 5.4 dB for clicks and 8 kHz pure tones, respectively. There was no significant effect of the noise trauma on the mean ABR thresholds to 2 kHz and 4 kHz pure tones.

The ABR measurements to clicks were investigated in more detail with regard to their ABR wave I and IV amplitudes, latencies, slopes of the wave I and IV amplitude growth functions and the wave IV/I amplitude ratio in Fig 2. We found a significant effect of the noise trauma on the P1-N1 amplitude at 90 dB SPL with lower amplitudes two days post-trauma

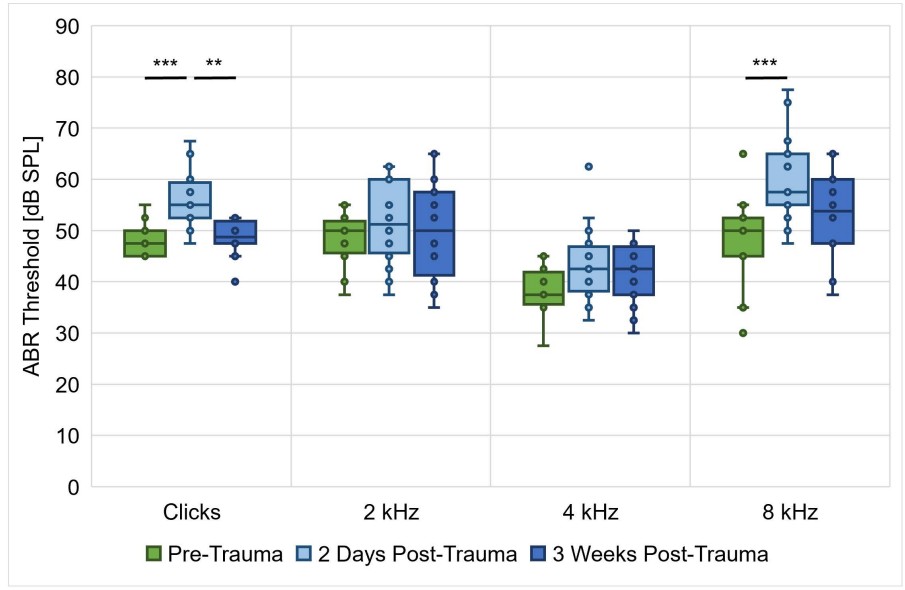

**Fig 1. ABR thresholds at three different time points.** ABR measurements to clicks and pure tones at 2, 4 and 8 kHz were done in all gerbils ($n = 14$) immediately before, two days after and three weeks after noise exposure. ABR thresholds to clicks were significantly elevated two days after noise exposure but were recovered three weeks post-trauma. For pure tones at 2 and 4 kHz, there were no significant differences in ABR thresholds before and after noise trauma. However, ABR thresholds to 8 kHz pure tones were significantly elevated 2 days post-trauma and were not clearly recovered by three weeks post-exposure. **: $p < 0.01$, ***: $p < 0.001$.

and three weeks post-trauma compared to pre-trauma (Friedman-test: $\chi^2(2, 14) = 8.714$, $p = 0.013$; Dunn-Bonferroni post-hoc tests: pre-trauma vs. two days post-trauma, $z = 1.000$, $p = 0.024$; pre-trauma vs. three weeks post-trauma, $z = -0.929$, $p = 0.042$; Fig 2A), indicating a permanent negative effect of the noise trauma on the amplitude of ABR wave I. The latency of P1 at 90 dB SPL was not affected by noise exposure (Fig 2B), but there was a significant reduction in slope of the P1-N1 amplitude growth function for 60–90 dB SPL three weeks post-trauma in comparison to pre-trauma (Friedman-test: $\chi^2(2, 14) = 9.000$, $p = 0.011$; Dunn-Bonferroni post-hoc tests: pre-trauma vs. three weeks post-trauma, $z = 1.071$, $p = 0.014$; Fig 2C), which is in line with the permanent noise trauma effect on the amplitude of ABR wave I. ABR wave IV, however, was not affected by the noise exposure, since neither P4-N4 amplitude at 90 dB SPL (Fig 2D), P4 latency at 90 dB SPL (Fig 2E) nor the slope of the P4-N4 amplitude growth function for 60–90 dB SPL (Fig 2F) were affected by the acoustic trauma. As a consequence of the reduction in P1-N1 amplitude and a lack of change in P4-N4 amplitude, the (P4-N4)/(P1-N1) amplitude ratio was significantly elevated two days after noise exposure compared to pre noise exposure (Friedman-test: $\chi^2(2, 14) = 6.143$, $p = 0.046$; Dunn-Bonferroni post-hoc tests: pre-trauma vs. two days post-trauma, $z = -0.929$, $p = 0.042$; Fig 2G).

In order to investigate potential noise-induced changes in the quantity of synapses between IHCs and ANFs, synapses were counted for a subset of the acoustically traumatized gerbils ($n = 9$). Fig 3 shows the gerbils' synapse counts at cochlear positions equivalent to 1, 2, 4, 8 and 16 kHz. The data was compared to synapse counts of young-adult gerbils from a previous study ($n = 7$) that were not acoustically traumatized [39]. The results showed a significant main effect of cochlear position and noise trauma on the synapse counts, but no significant interaction effect between cochlear position and noise trauma (repeated-measures ANOVA, factor cochlear position: $F(4, 56) = 6.318$, $p < 0.001$, factor noise trauma: $F(1, 14) = 5.041$, $p = 0.041$, cochlear position x noise trauma: $F(4, 56) = 0.343$, $p = 0.848$; Fig 3). Bonferroni-corrected post-hoc testing showed that the observed main effect of cochlear position was based on a significantly larger

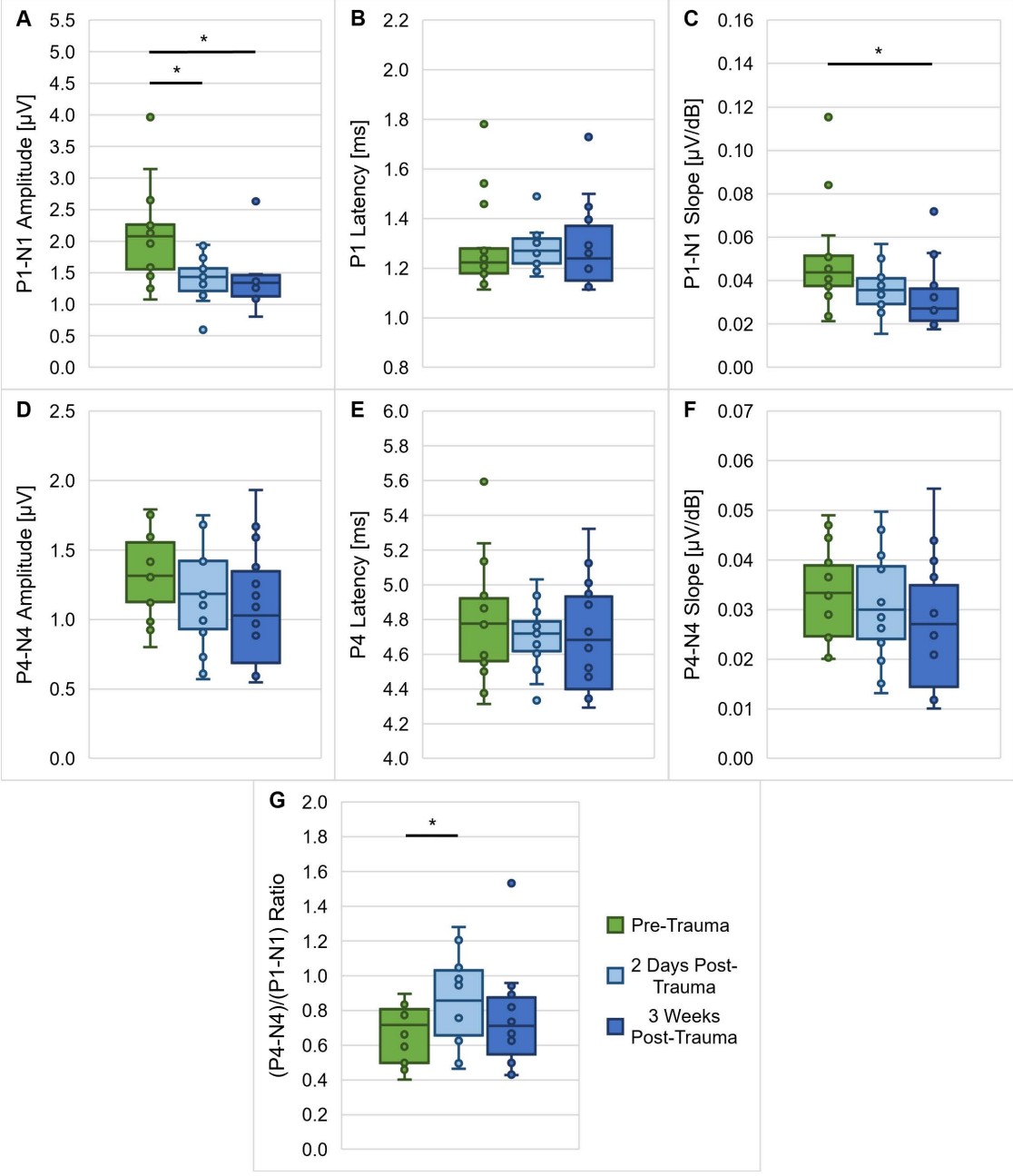

**Fig 2. ABR wave I and IV parameters in response to clicks.** The ABR measurements to clicks that were done in all gerbils ($n = 14$) immediately before, two days after and three weeks after noise exposure (see Fig 1), were investigated in more detail with regard to their ABR wave I and IV amplitudes, latencies, slopes of the wave I and IV amplitude growth functions and wave IV/I amplitude ratio. The P1-N1 amplitude at 90 dB SPL (**A**) was significantly reduced two days post-trauma compared to pre-trauma and stayed decreased three weeks post-trauma. The slope of the P1-N1 amplitude growth function for 60–90 dB SPL (**C**) was significantly lower three weeks after noise exposure compared to before noise exposure. No differences were found in P1 latency at 90 dB SPL (**B**), as well as P4-N4 amplitude at 90 dB SPL (**D**), P4 latency at 90 dB SPL (**E**) and the slope of the P4-N4 amplitude growth function for 60 – 90dB SPL (**F**). The (P4-N4)/(P1-N1) amplitude ratio (**G**) was significantly elevated two days after noise exposure. *: $p < 0.05$.

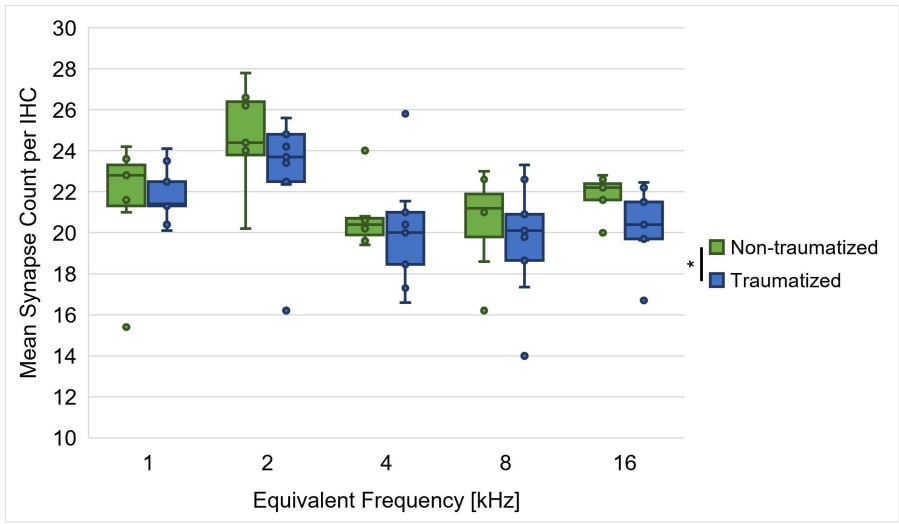

**Fig 3. Synapse counts of acoustically traumatized and non-traumatized gerbils at five cochlear positions.** Synapse counts for a subset of the acoustically traumatized gerbils ($n = 9$) were assessed at five different cochlear positions (equivalent to 1, 2, 4, 8 and 16 kHz) and compared to synapse counts of young-adult gerbils that were not acoustically traumatized from a previous study ($n = 7$) [39]. Synapse counts were significantly lower in acoustically traumatized compared to non-traumatized gerbils. Larger numbers of synapses were found at the cochlear position equivalent to 2 kHz in comparison to the cochlear positions equivalent to 4, 8 and 16 kHz for both groups of gerbils. *: $p < 0.05$.

number of synapses for the cochlear position equivalent to 2 kHz, compared to the cochlear positions equivalent to 4 kHz ($M_{Diff} = 3.538$, 95%-CI[0.011, 7.066], $p = 0.049$), 8 kHz ($M_{Diff} = 3.785$, 95%-CI[0.256, 3.315], $p = 0.031$) and 16 kHz ($M_{Diff} = 2.798$, 95%-CI[0.334, 5.261], $p = 0.020$). Most importantly, the main effect of noise trauma on the synapse counts was found to be due to a significantly lower number of synapses in the acoustically traumatized gerbils compared to the non-traumatized gerbils.

### Overall behavioral vowel discrimination performance improved after acoustic trauma despite temporary noise-induced hearing loss

The gerbils' vowel and consonant discrimination abilities were tested in different CVC and VCV conditions, respectively, with varying phonetic contexts. In a previous study [33] we saw that the phonetic contexts in the different CVC and VCV conditions were not affecting the gerbils' overall $d'$-values and response latencies for the discrimination of the medial vowels and consonants in the logatomes, respectively. Because of this, the results from the different CVC or VCV conditions of each gerbil ($n = 9$) were pooled for each of the time points (pre-trauma and post-trauma) enabling joined analyses of all CVC and VCV conditions, respectively. We then tested for general differences between pre-trauma and post-trauma datasets, also with respect to the logatome type (CVC vs. VCV). The results indicated no main effect of the noise trauma on the mean $d'$-value, but a main effect of the logatome type (higher $d'$-values for CVCs than for VCVs) and an interaction effect between noise trauma and logatome type (repeated-measures ANOVA, factor logatome type: $F(1, 7) = 244.788$, $p < 0.001$, noise trauma x logatome type: $F(1, 7) = 20.041$, $p = 0.003$; Fig 4A). The interaction effect was traced back to a significant increase in $d'$-values for CVCs after noise exposure in comparison to before noise exposure ($M_{Diff} = -0.273$, 95%-CI[−0.458, −0.088], $p = 0.010$), which was absent for VCVs. Further, there were main effects of noise exposure and logatome type on the response latencies (repeated-measures ANOVA, factor noise trauma: $F(1, 7) = 18.678$, $p = 0.003$, factor logatome type: $F(1, 7) = 117.174$, $p < 0.001$; Fig 4B), with shorter response latencies post-exposure compared to pre-exposure and for CVCs compared to VCVs, respectively. More importantly, these factors also showed an interaction

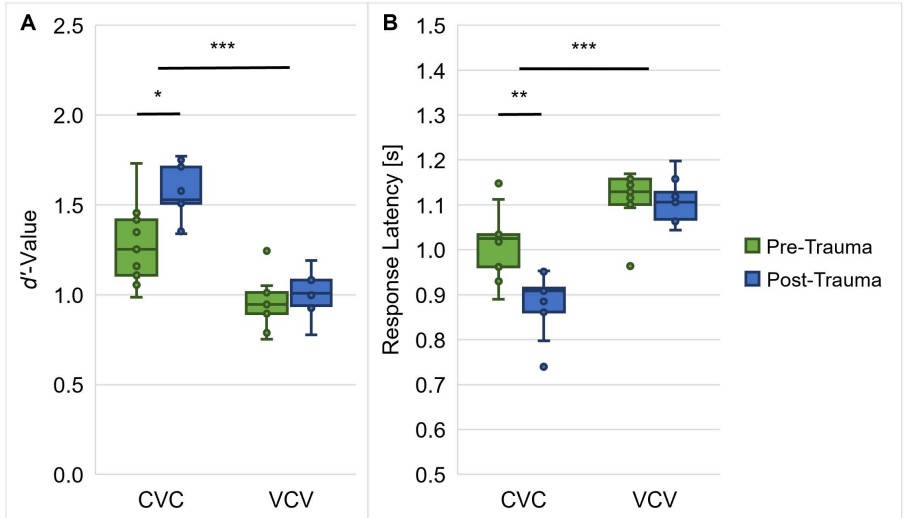

**Fig 4. Influence of acoustic trauma on overall speech sound discrimination ability.** *d'*-values (**A**) for the behavioral performance of the gerbils (*n* = 9) in the CVC conditions were significantly higher than in the VCV conditions pre- and post-trauma. Response latencies (**B**) before and after trauma were significantly longer for the discrimination of consonants in comparison to the discrimination of vowels. In addition, trauma effects were found for response latencies with shorter response latencies post-trauma compared to pre-trauma, while there was no main effect of trauma on the *d'*-values. There were interaction effects between trauma and logatome type on *d'*-value and response latency. Specifically, the gerbils showed significantly shorter response latencies and significantly higher *d'*-values for CVC conditions (but not for VCV conditions) post-trauma compared to pre-trauma. *: $p < 0.05$, **: $p < 0.01$, ***: $p < 0.001$.

effect on the response latencies (repeated-measures ANOVA, noise trauma x logatome type: $F(1, 7) = 13.771$, $p = 0.008$; Fig 4B). This interaction effect was found to be due to shorter response latencies for CVCs post-trauma compared to pre-trauma ($M_{Diff} = 0.129$, 95%-CI[0.064, 0.195], $p = 0.002$), while there was no difference in response latencies for VCVs pre-trauma vs. post-trauma. For the mean hit rate we found the same pattern as for the mean *d'*-value with significantly higher rates for CVC conditions (but not for VCV conditions) post-trauma compared to pre-trauma, while the false alarm rate was neither effected by the noise trauma nor by the logatome type (S1 Fig).

## Noise exposure did not change general organization of vowels and consonants in perceptual maps

The behavioral paradigm in the present study was used to investigate the gerbils' ability to discriminate vowels and consonants before and after a noise trauma. With the help of MDS, perceptual maps were generated in which long distances indicate a good discriminability between phonemes, while short distances correspond to a poor discriminability.

Fig 5A and 5B show the two-dimensional perceptual maps for vowels that integrate data from all CVC conditions pre-trauma and post-trauma, respectively. The vowels in the pre- and post-trauma perceptual maps were arranged in the same way and this constellation was very similar to the vowel chart of Northern Standard German (Fig 5C). In the vowel chart the position of the vowels is determined by their articulatory features tongue height and tongue backness. The articulatory configuration then again determines the frequency of the first (F1) and second (F2) formant of the vowels. Thus, the F1 frequency highly correlates with the Dimension 2 coordinates ($R^2 = 0.945$ and 0.959 for pre- and post-trauma, respectively) and the F2 frequency highly correlates with the Dimension 1 coordinates of the vowels in the perceptual maps ($R^2 = 0.855$ and 0.897 for pre- and post-trauma, respectively). Property vectors that are based on linear regressions of the vowel coordinates and the frequency of F1 and F2 were plotted as blue and red arrows in the perceptual maps, respectively (Fig 5A and 5B). Further, the perceptual maps had a very good fit to the underlying pre- and post-trauma data, which was quantified by DAF values that amounted to 0.932 and 0.951, respectively.

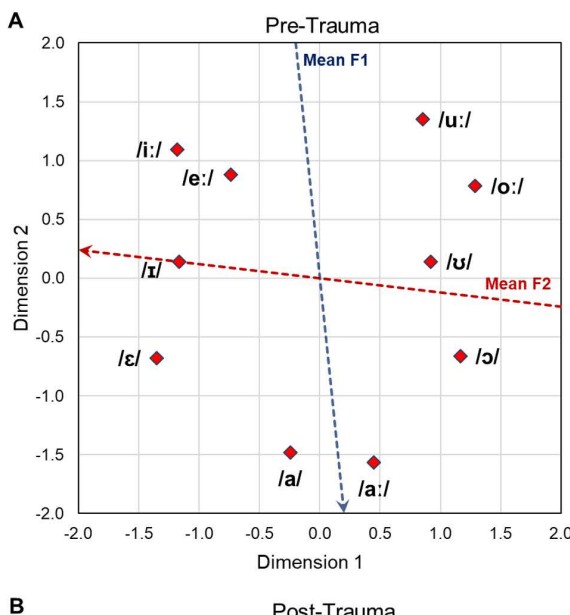

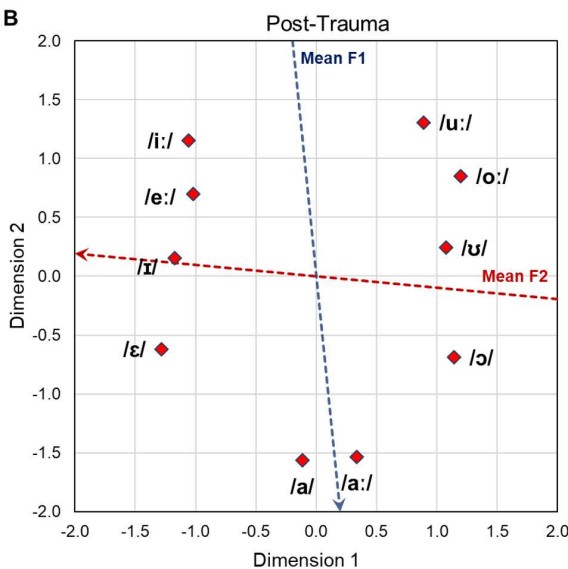

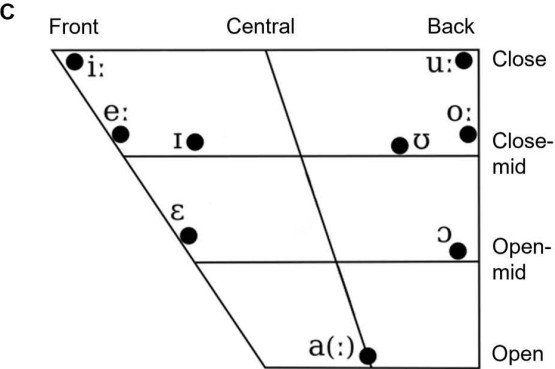

**Fig 5. Two-dimensional perceptual maps for vowels.** Two-dimensional perceptual maps for vowels were generated integrating the data from all CVC conditions of all pre-trauma (**A**) and post-trauma (**B**) datasets, respectively. Overall arrangement as well as individual locations of the vowels are very similar for the pre-trauma and post-trauma datasets and resemble the arrangement in the vowel chart for Northern Standard German (**C**, edited

from [45]). In the vowel chart, the vowels are organized according to their articulatory features tongue backness (front – back) and tongue height (open – close). The tongue backness during articulation determines the frequency of F2, while changes in the tongue height lead to different F1 frequencies. Additionally, the vowels in the perceptual maps were also arranged according to the frequencies of their first two formants. The red and blue dotted arrows in **A** and **B** show the axes along which the mean frequencies of the second (F2) and first (F1) formant increase, respectively.

The three-dimensional perceptual maps for consonants integrating the data from all VCV conditions from before and after the noise trauma are shown in Fig 6A and 6B, respectively. As for the CVCs, also the gerbils' overall perception of the VCVs was generally similar before and after noise exposure. The similarity was quantified by calculating the average squared distance between corresponding consonants post-trauma versus pre-trauma, after Procrustes rotation and translation, which was only 0.8% of the average squared distance between all consonants within each trauma condition. The consonants in both perceptual maps were clustered according to their articulatory features manner of articulation, place of articulation and voicing, which are indicated by color, shape, and border of the respective consonant symbol. The three perspectives on each of the perceptual maps were chosen to point out the organization following these articulatory features. The DAF values for the perceptual maps amounted to 0.944 and 0.943 for pre- and post-trauma, respectively, indicating a very good fit to the underlying data.

In a next step, we plotted the average response latencies between all vowel and consonant pairs for post-trauma data vs. pre-trauma data in Fig 7A and 7B, respectively. Corresponding to the high similarity of the perceptual maps before and after noise exposure, the pre- and post-trauma response latencies between the vowel pairs were highly and significantly correlated ($r_s$(43) = 0.964, $p < 0.001$; Fig 7A). The response latencies between the consonant pairs before and after noise trauma were less, but still highly and significantly correlated ($r_s$(64) = 0.905, $p < 0.001$, Fig 7B). Noteworthy, the response latencies for the discrimination of CVCs but not VCVs were shifted towards shorter latencies in the post-trauma data compared to the pre-trauma data, as already seen in Fig 4B. While the response latencies for the consonant discriminations were situated along the identity line (formula for linear regression of VCV response latencies: *y = 1.039x - 0.057*), the response latencies for the vowel discriminations were aligned to a regression line that was deviating most from the identity line for relatively short latencies and that crossed the identity line for long latencies (formula for linear regression of CVC response latencies: *y = 1.314x - 0.455*).

The lower correlation of the response latencies in the consonant discrimination compared to the vowel discrimination is not due to the noise exposure, but there was generally a higher inter-individual variability in the discrimination of consonants compared to vowels. We investigated the inter-individual variability by calculating Spearman's rank correlations between the response latencies of all individual gerbils ($n = 9$) pre- and post-trauma and determined the mean correlations for CVCs and VCVs of all subjects (Fig 8). We found significant main effects of logatome type and noise trauma with generally larger mean Spearman's rank correlations of the response latencies for CVCs compared to VCVs and post-trauma compared to pre-trauma (repeated-measures ANOVA, factor logatome type: $F(1, 7) = 63.872$, $p < 0.001$, factor noise trauma: $F(1, 7) = 78.897$, $p < 0.001$). There was no significant interaction between logatome type and noise trauma. Thus, the inter-individual variability was significantly higher for the discrimination of consonants than for the discrimination of vowels, and it was significantly reduced after the noise overexposure compared to before the noise overexposure. The correlations were higher post-trauma compared to pre-trauma for both vowels and consonants, even though the larger inter-individual consistency in response latencies did not lead to an increase in overall consonant discrimination ability.

## Noise trauma led to further improvement of most salient vowel discriminations

The different articulatory features of vowels and consonants and their influence on the response latencies between phoneme pairs were investigated in greater detail with regard to possible noise-induced changes. Mean response latencies of each animal ($n = 9$) for all vowel pairs with a specific combination of characteristics for the different articulatory features were calculated and compared between pre-trauma and post-trauma data. For the discriminations between vowels,

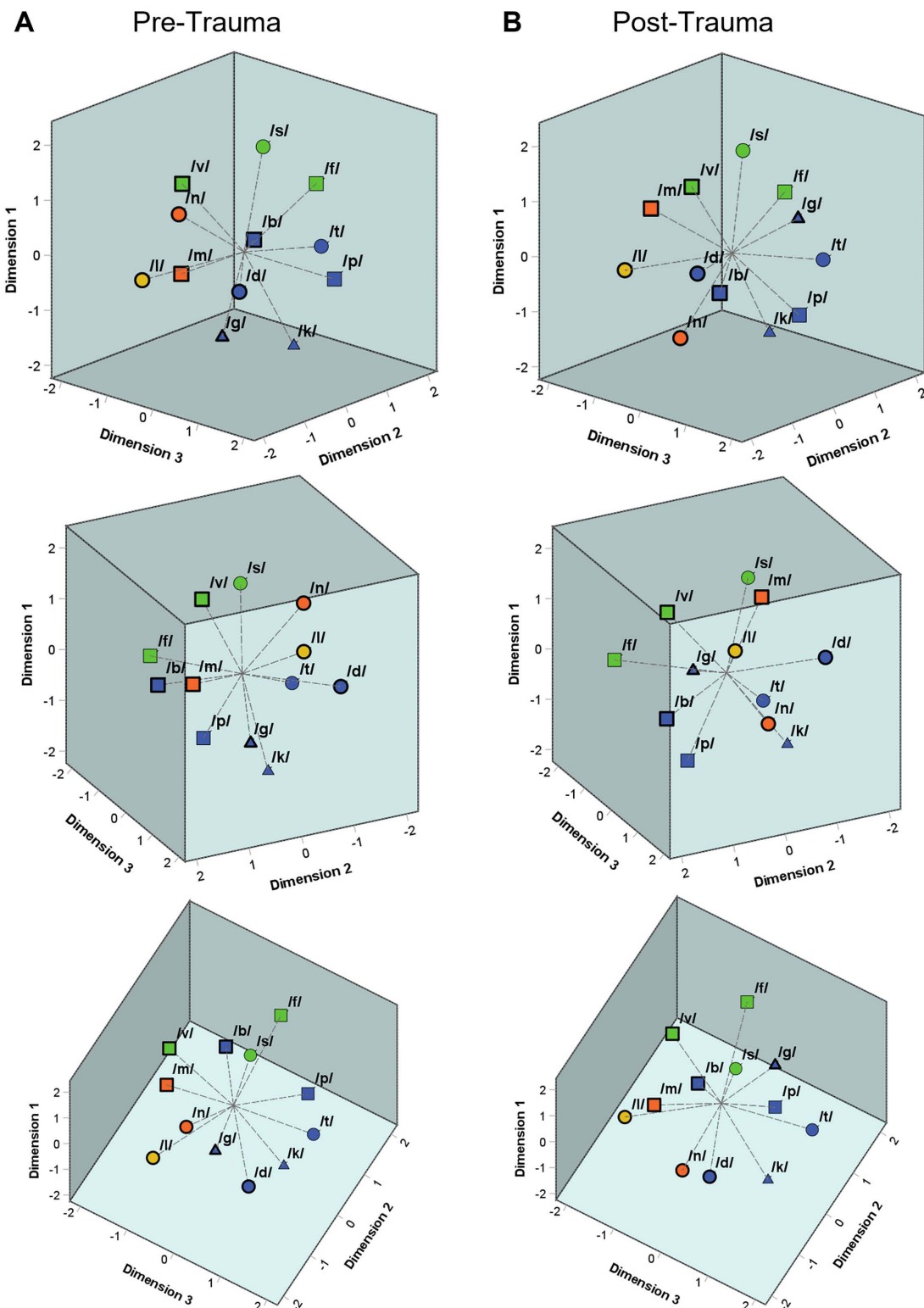

**Fig 6. Three-dimensional perceptual maps for consonants.** Three-dimensional perceptual maps for consonants were generated integrating the data from all VCV conditions of all pre-trauma (**A**) and post-trauma (**B**) datasets, respectively. Both perceptual maps are shown from three different perspectives enabling an easier visualization of the three-dimensional arrangement of the consonants. The consonants were clustered according to their

articulatory features both before and after noise exposure. The different manners of articulation can be differentiated by color (blue = plosive, green = fricative, orange = nasal, yellow = lateral approximant). The place of articulation is indicated by shape (□ = labial, O = coronal, △ = dorsal). The different voicing characteristics are marked by border (**thick border = voiced**, thin border = voiceless). Depending on the perspective, one can see that the consonants were clustered according to different characteristics of all of these articulatory features (manner of articulation in the left panels, place of articulation in the central panels and voicing in the right panels) both before noise trauma and after noise trauma.

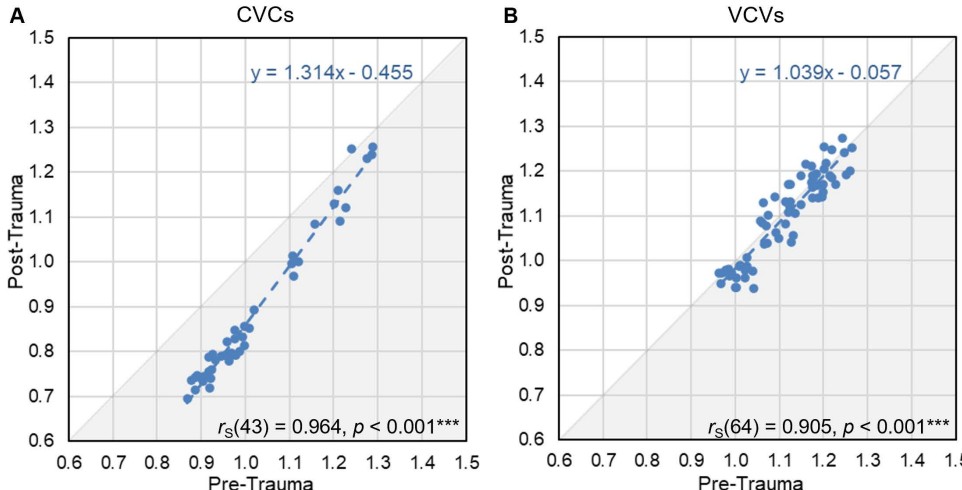

**Fig 7. Correlation between pre- and post-trauma response latencies.** The mean response latencies pooled across all gerbils of all vowel discriminations (each dot corresponds to one vowel pair) (**A**) for pre-trauma vs. post-trauma were highly and significantly correlated ($r_S(43) = 0.964$, $p < 0.001$***; formula for linear regression of CVC response latencies: $y = 1.314x - 0.455$). The correlation of the mean response latencies pooled across all gerbils of all consonant discriminations (each dot corresponds to one consonant pair) (**B**) for pre-trauma vs. post-trauma was also highly significant, however, there was a weaker linear relationship compared to the vowels ($r_S(64) = 0.905$, $p < 0.001$***; formula for linear regression of VCV response latencies: $y = 1.039x - 0.057$). For CVCs, the response latencies were shifted towards shorter latencies post-trauma compared to pre-trauma.

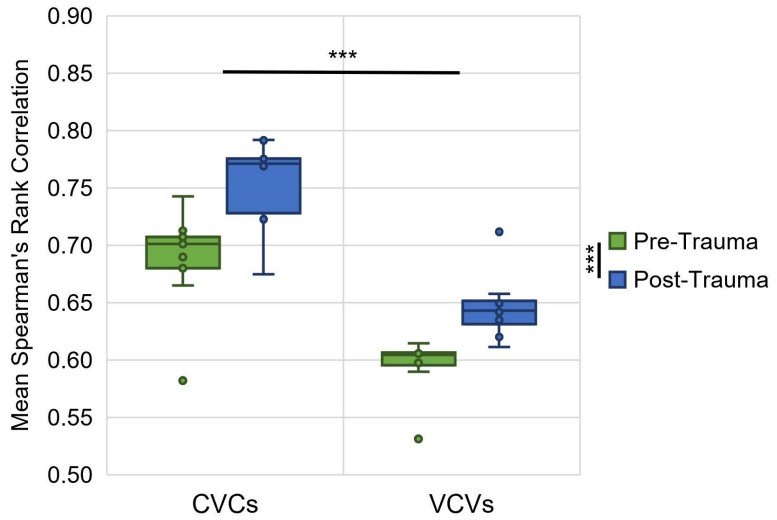

**Fig 8. Spearman's rank correlations of individual pre- and post-trauma response latencies.** Spearman's rank correlations were calculated between the response latencies of pre- and post-trauma datasets of all individual gerbils ($n = 9$) for CVCs and VCVs. Correlations between the gerbils were generally higher for CVCs than for VCVs, independent of the trauma. For both CVCs and VCVs, correlations among post-trauma gerbils were higher than correlations among pre-trauma gerbils. ***: $p < 0.001$.

the characteristics of the articulatory features tongue height (Fig 9A), tongue backness (Fig 9B) and information about which articulatory features the vowels have in common (Fig 9C) were considered for the analysis. The results indicated significant main effects of tongue height and noise trauma on the response latencies as well as an interaction (repeated-measures ANOVA, factor tongue height: $F(3.114, 24.915) = 131.529$, $p < 0.001$, factor noise trauma: $F(1, 8) = 25.055$,

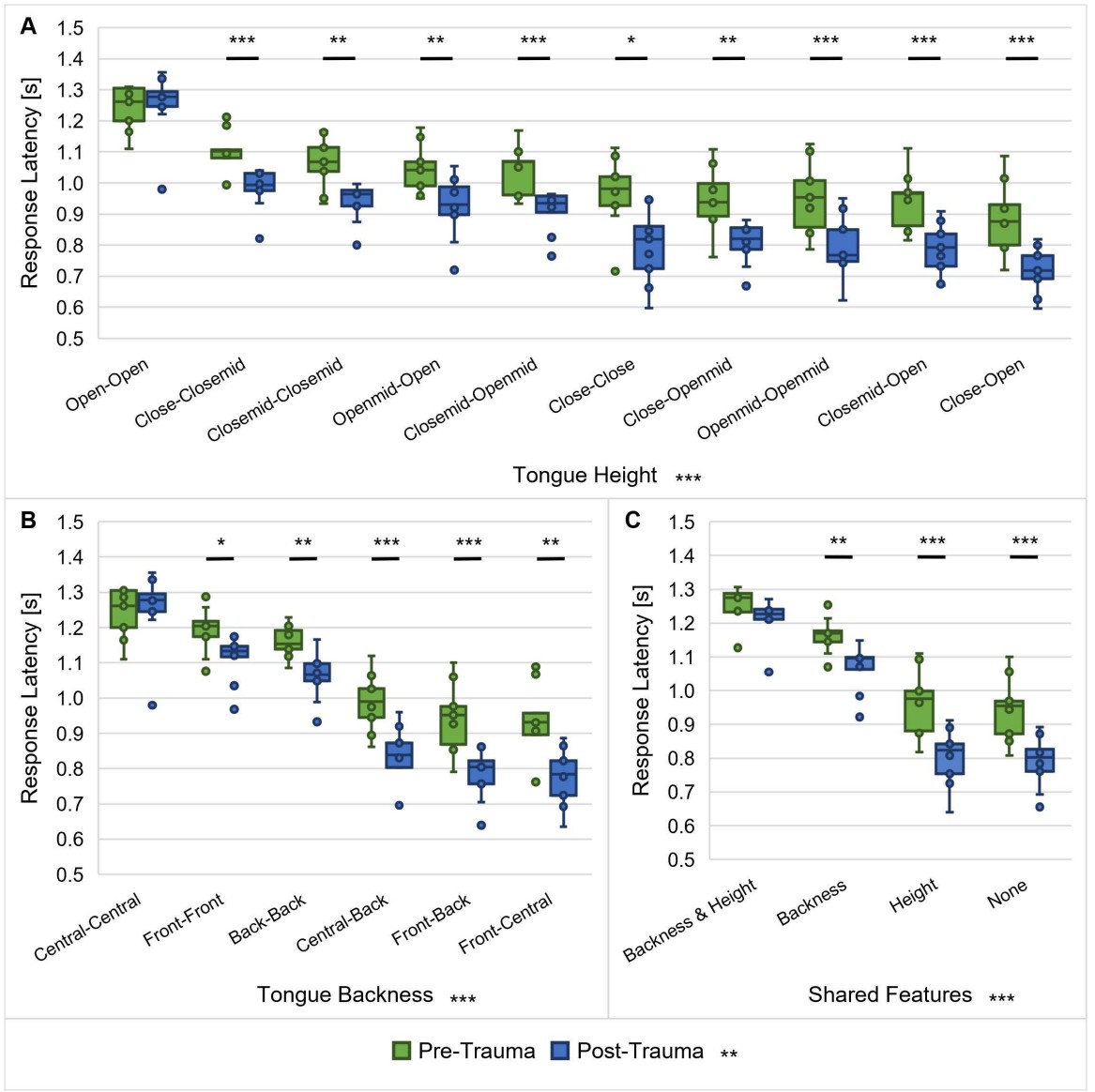

**Fig 9. Response latencies for discriminations of different vowel types.** Response latencies of all gerbils ($n = 9$) between vowel pairs pre- and post-trauma were investigated with regard to their tongue height (**A**), tongue backness (**B**) and shared articulatory features (**C**). Response latencies were found to be significantly different between various combinations of tongue heights, tongue backness and shared articulatory features. There was a significant main effect of trauma on the response latencies for all of the articulatory features, with shorter response latencies post-trauma compared to pre-trauma. Moreover, there were interaction effects between trauma and each of the articulatory features resulting in the most distinct shortening of the response latencies post-trauma for those vowel discriminations that already showed the shortest response latencies before the noise exposure, while there was no trauma effect on the response latencies of those vowel discriminations that had the longest response latencies. *: $p < 0.05$, **: $p < 0.01$, ***: $p < 0.001$.

*p* = 0.001, tongue height x noise trauma: $F(3.243, 25.948) = 7.919$, $p < 0.001$; Fig 9A). Bonferroni-corrected post-hoc tests showed that there were significant differences in the response latencies between the majority (78%) of the different combinations of tongue heights and the averaged response latencies for the different combinations of tongue heights were overall shorter post-trauma compared to pre-trauma. The interaction effect between trauma and tongue height is reflected in a differential effect of the noise exposure on the response latencies, which depends on the tongue heights of the vowels. While there was no effect of the noise trauma on the response latencies for the discrimination of vowels with open tongue height configurations, there were significant, but varyingly large effects on the response latencies for vowel discriminations with other tongue height configurations. Specifically, there tended to be the largest differences in response latencies pre-trauma vs. post-trauma for vowel discriminations with short latencies, while there were smaller differences for vowel discriminations with long latencies.

Further, we found significant differences in response latencies for the different combinations of tongue backness and between pre-trauma and post-trauma data, as well as an interaction effect between tongue backness and noise trauma (repeated-measures ANOVA, factor tongue backness: $F(5, 40) = 238.609$, $p < 0.001$, factor noise trauma: $F(1, 8) = 23.719$, $p = 0.001$, tongue backness x noise trauma: $F(5, 40) = 11.641$, $p < 0.001$; Fig 9B). All except for one of the pairwise comparisons (front – central vs. front – back) in the Bonferroni-corrected post-hoc testing between the different combinations of tongue backness were significant and the mean response latencies for the different combinations of tongue backness were again shorter post-trauma in comparison to pre-trauma. Most importantly, the interaction effect between noise trauma and tongue backness was found to be caused by a differential effect of the noise exposure on the response latencies, depending on the tongue backness of the vowels. In more detail, there was no difference pre-trauma vs. post-trauma in response latencies for discriminations of two centrally articulated vowels, while there were significant differences for discriminations between two front vowels or two back vowels and the largest differences between pre- and post-trauma were found for discriminations of vowels with different tongue backness. Generally, the reduction in response latency post-trauma compared to pre-trauma was most pronounced for those vowel discriminations that already showed the shortest response latencies before the noise exposure, while there was less or no shortening of the response latencies for vowel discriminations that showed rather long response latencies before the noise overexposure. In other words, the easier the vowel discrimination the more the discriminability was enhanced by the noise exposure.

Finally, also for the shared articulatory features a significant effect on the response latencies was observed, together with a main effect of the noise trauma and an interaction effect between the shared articulatory features and the noise trauma (repeated-measures ANOVA, factor shared features: $F(1.660, 13.279) = 324.233$ $p < 0.001$, factor noise trauma: $F(1, 8) = 24.212$, $p = 0.001$, shared features x noise trauma: $F(1.286, 10.285) = 22.154$, $p < 0.001$; Fig 9C). Bonferroni-corrected post-hoc tests showed that the response latencies were significantly different between all combinations of shared articulatory features and that the averaged response latencies for the different combinations of shared articulatory features were shorter post-trauma compared to pre-trauma. The interaction effect was due to a differential effect of the noise exposure on the response latencies, depending on the shared articulatory features of the vowels. As for the tongue height and tongue backness, vowel discriminations with combinations of articulatory features that showed rather short response latencies before the noise exposure tended towards even shorter response latencies after the noise exposure. This means that those vowels that already showed a good discriminability before the noise trauma were perceived as even more distinct after the noise trauma. Vowel discriminations with rather long response latencies before the noise exposure were perceived as difficult as before the noise exposure or only improved marginally after the noise trauma, though.

## Overall consonant discrimination remains largely unchanged by noise exposure

For the discrimination of consonants, the articulatory features manner of articulation (Fig 10A), voicing (Fig 10B), place of articulation (Fig 10C) and the shared articulatory features (Fig 10D) were investigated with regard to their influence on the

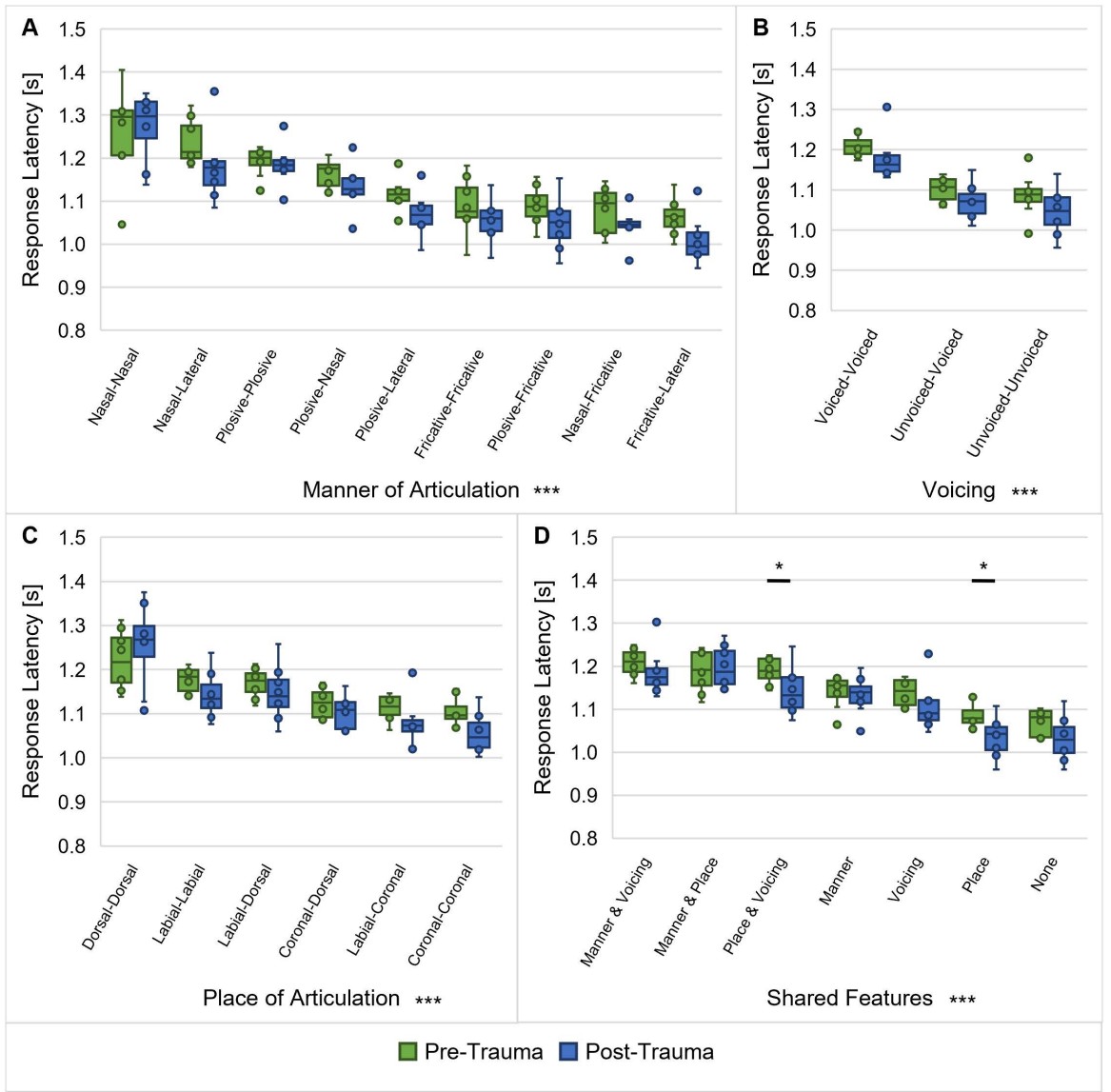

**Fig 10. Response latencies for discriminations of different consonant types.** Response latencies between consonant pairs from before and after noise exposure were investigated for all gerbils ($n = 9$) with regard to their manner of articulation (**A**), voicing (**B**), place of articulation (**C**) and shared articulatory features (**D**). Response latencies were found to be significantly different between various combinations of manners of articulation, voicing, places of articulation and shared articulatory features. There was no main effect of trauma on the response latencies for any of the articulatory features. However, there was an interaction effect of trauma and the shared articulatory features with shorter response latencies post-trauma compared to pre-trauma between consonants that shared their place of articulation or that had both the same the place of articulation and the same voicing. *: $p < 0.05$, ***: $p < 0.001$.

response latencies of all gerbils ($n = 9$) pre- and post-trauma. The results indicated a main effect of the manner of articulation on the response latencies, independent of noise exposure and without an interaction between manner of articulation and noise trauma (repeated-measures ANOVA, factor manner of articulation: $F(1.871, 13.100) = 39.014$, $p < 0.001$; Fig 10A). As revealed by Bonferroni-corrected post-hoc tests, half of the pairwise comparisons (50%) between the different combinations of manners of articulation had significantly different response latencies.

The articulatory feature voicing also showed a main effect on the response latencies, with longer latencies for consonant discriminations between two voiced consonants compared to consonant discriminations between two voiceless or one voiced and one voiceless consonant, while there was no effect of the noise exposure and no interaction between voicing and noise trauma (repeated-measures ANOVA, factor voicing: $F(1.072, 7.502) = 63.885$, $p < 0.001$; Fig 10B).

Additionally, we found that different constellations of places of articulation led to significantly different response latencies and that there was no main effect of noise exposure and no interaction effect between the place of articulation and noise exposure (repeated-measures ANOVA, factor place of articulation: $F(1.369, 9.581) = 37.915$, $p < 0.001$; Fig 10C). Bonferroni-corrected post-hoc testing showed that the response latencies were significantly different between the majority (67%) of the different combinations of places of articulation.

Finally, response latencies differed significantly depending on the articulatory features that the consonant pairs share, without a main effect of noise exposure, but with an interaction effect between shared features and the noise trauma (repeated-measures ANOVA, factor shared features: $F(6, 42) = 110.195$, $p < 0.001$, shared features x noise trauma: $F(6, 42) = 2.688$, $p = 0.027$; Fig 10D). Bonferroni-corrected post-hoc testing revealed that the response latencies were significantly different for the majority (76%) of the pairwise comparisons of the different combinations of shared articulatory features. Further, the response latencies for discriminations between consonants with the same place of articulation and consonants with both the same place of articulation and the same voicing was significantly shorter after the noise exposure in comparison to before the noise exposure.

## Discussion

Hearing loss due to noise overexposure is a widespread problem in the industrialized world. For the research on NIHL, the usage of animal models can be very valuable, since they allow for a controlled noise exposure in subjects that can serve as their own controls, leading to a good interpretability of the data in contrast to the restricted methodological possibilities in studies with human subjects who typically show large differences in noise exposure history. In the present study, we evaluated the effects of noise overexposure on the speech sound discrimination ability of Mongolian gerbils – a widely used animal model for NIHL [e.g., 13,21–25] – in order to investigate the commonly suggested link between noise overexposure and speech-in-noise perception difficulties.

### NIHL with temporary threshold shift and synaptopathy apparent in ABRs of noise-exposed gerbils

The acoustic trauma in the present study caused an ABR threshold shift in response to clicks and pure tones at 8 kHz in the exposed gerbils (Fig 1). This matches the expectations based on results from previous studies [e.g., 13,21,22] and the known cochlear nonlinearities [46], leading to the largest noise trauma effects at frequencies above the exposure noise band (2–4 kHz). In the acute phase of the acoustic trauma (2 days post-trauma), the mean ABR thresholds for both clicks and 8 kHz tones were significantly increased by on average 7.7 dB and 12.1 dB, respectively. Three weeks post-trauma, the remaining – presumably permanent – threshold shifts amounted on average to 0.4 dB and 5.4 dB for clicks and 8 kHz pure tones, respectively, and were not significantly different from pre-trauma thresholds. Thus, the here applied noise overexposure caused a temporary threshold shift (TTS). In comparison to previous studies, the magnitude of the observed threshold shift is rather small. The majority of previous noise exposure studies in gerbils with high exposure levels (107–130 dB SPL) reported a permanent threshold shift [e.g., 21,47–49] in contrast to studies with lower exposure levels (80–105 dB SPL) mostly reporting a TTS [e.g., 50,51]. However, it is very difficult to compare the noise trauma effects of different studies, since the exposure levels, frequencies, durations and setups vary largely between studies. Further, in some studies gerbils were anesthetized for the noise overexposure, while in other studies, gerbils were fully awake during acoustic traumatization. Also, individuals of various ages were used in the different studies. Consequently, these differences in the trauma conditions may contribute to differential effects of noise overexposure on the ABRs of gerbils. In our case, especially the exposure to an octave-band noise (in contrast to narrow-band noise bands or pure tones) [52]

and the use of anesthesia during noise overexposure may have preserved the gerbils from larger threshold shifts, as it has been shown that anesthesia can have protective effects on hair cells [53] and hearing thresholds [54]. Such protective effects might be due to intrinsic properties of anesthetics or a lack of stress processes that are activated in awake animals during noise overexposure [55].

In contrast to the recovered thresholds, the amplitude of ABR wave I and the slope of its growth function were significantly decreased three weeks post-exposure compared to pre-exposure (Fig 2A and 2C), which is in line with observations from previous acoustic trauma studies in gerbils [13,22,52]. Such a persistent reduction in the amplitude of ABR wave I, which originates mainly from activity of the auditory nerve [56], might indicate noise-induced synaptopathy, even when there is no permanent change in the hearing thresholds of the exposed gerbils. Indeed, studies in rodents found that hearing thresholds can stay in a normal range despite of up to 60% losses of synapses between IHCs and ANFs [15,57]. As an explanation for the recovery of auditory thresholds despite severe noise-induced neuropathy, it was hypothesized that noise overexposure may lead to a selective loss of low- and medium-SR ANFs [12,13]. In this way, an acoustic trauma could result in normal (completely recovered) cochlear thresholds and no or only minimal damage of hair cells in combination with a persistent loss of ANFs [15]. Such a loss of low- and medium-SR ANFs might play a role in the speech-in-noise perception difficulties of listeners with NIHL [11], since low-SR fibers typically have high thresholds and are thought to be particularly important for robust signal coding of complex sounds in the presence of background noise [58,59]. In line with this hypothesis about a connection between noise exposure and synaptopathy, we found significantly lower synapse counts for acoustically traumatized gerbils in comparison to non-traumatized gerbils (Fig 3). Thus, we can confirm a direct connection between noise overexposure and a loss of synapses in our gerbil model for NIHL. It is important to note that besides a loss of whole synapses, also differences in the morphology of synapses, e.g., with an increased number of pre-synaptic ribbons missing their associated post-synaptic glutamate receptor patches, and changes in the synaptic position along the IHC's membrane have previously been observed after acoustic traumatization [60,61]. In line with this, we also observed abnormalities in synaptic location in the noise-exposed gerbils with more synapses being located in an uncommon position towards the top of the IHC than typically observed in gerbils that were not noise-exposed, which may have further functional consequences. Thus, the significantly reduced synapse numbers together with changes in synaptic properties are probably responsible for the observed permanent reductions in ABR wave I amplitude.

In summary, even though the ABR thresholds of the noise-exposed gerbils were not significantly different three weeks post-exposure compared to pre-exposure, the persistent reductions in amplitude of ABR wave I and the slope of its growth function as well as the significant reduction in synapse numbers after acoustic traumatization suggest that the gerbils suffered from NIHL and synaptopathy.

### Enhanced behavioral vowel discrimination ability presumably caused by noise-induced physiological changes

In the behavioral experiments, we investigated how the discriminability of different types of vowels and consonants changed from before to after noise overexposure. Generally, we saw that the gerbils were better in discriminating vowels than consonants (higher *d'*-values and shorter response latencies; Fig 4), which agrees with what we observed previously in young normal-hearing gerbils that were never exposed to noise [31]. Further, we found large similarities in the pre- and post-trauma perceptual maps of both vowels and consonants. The overall arrangement and organization of the vowels (Fig 5) and consonants (Fig 6) in accordance with their articulatory features in the perceptual maps was similar before and after noise exposure, suggesting that generally the same cues were used for speech sound discrimination before and after noise exposure. However, the gerbils' overall behavioral discrimination ability for vowels – as assessed by *d'*-values – was increased after the noise overexposure compared to before the noise overexposure (Fig 4A). In line with this, the response latencies for vowel discriminations were significantly shorter post-trauma compared to pre-trauma (Fig 4B). This shortening in response latencies was not consistent across all vowel discriminations, but most pronounced for those discriminations that already showed comparatively short response latencies before the

noise exposure (Fig 7A). Consequently, the acoustic trauma seemed to affect the vowel discriminability differentially depending on the specific articulatory characteristics of the vowels (Fig 9), potentially changing the efficiency in using the articulatory differences (determining the frequency of F1 and F2) as a cue for discrimination. Thus, vowel discriminations with combinations of articulatory features that had the longest response latencies pre-trauma showed no or only small changes in their response latencies post-trauma, while vowel discriminations that were easy for the gerbils before the noise exposure showed an even lower difficulty after the noise exposure. The observed improvements in vowel discrimination ability correspond to findings from a study by Monaghan and colleagues (2020) that investigated neuronal responses to logatomes from the inferior colliculus (IC) and found that speech-in-noise discriminability at moderate sound intensities (60 dB SPL) was improved in gerbils four weeks after noise exposure (while it was declined at high intensities (75 dB SPL)) [51]. As a possible explanation for the level-dependent improved speech-in-noise discriminability after noise overexposure, a combination of peripheral synaptopathy, mainly targeting low- and medium-SR ANFs [12], together with a hypersensitivity due to a compensatory overall increase in central gain [62] was suggested [51]. A selective loss of low-SR ANFs with their high thresholds would particularly impair perception at high sound intensities, so that central gain changes might not fully compensate for the reduced input, while at moderate sound intensities the increase in central gain may even improve speech-in-noise discriminability. In line with this hypothesis about the increased central gain, average firing rates of IC neurons at moderate sound intensities were found to be significantly higher in noise-exposed gerbils compared to control gerbils [51]. Increases in central gain following noise overexposure or peripheral deafferentation have been reported for the auditory cortex and IC of different rodent species, and may be induced through synaptic disinhibition or an imbalance between excitatory and inhibitory responses in consequence of the reduced input [63–66]. These changes in central gain and the consequent hypersensitivity have been observed particularly for the tuning curve tails of noise-exposed ANFs and frequencies below the region of cochlear damage [67–70]. Such an overrepresentation of low-frequency stimulus energy comes along with a distorted tonotopy, which was found to be the dominant factor explaining changes in vowel- and consonant-coding in a chinchilla model of NIHL [71,72]. Besides hypersensitivity and a distorted tonotopy, also an enhanced envelope encoding might be a consequence of increased central gain following noise overexposure [73], potentially exerting a positive effect on vowel discriminability. In line with this, envelope encoding was found to be enhanced after noise overexposure in the auditory nerve and midbrain of acoustically traumatized chinchillas [74–76] as well as in evoked responses from the auditory cortex of human listeners with sensorineural hearing loss [77]. An increase in central gain together with the following hypersensitivity and enhancement in envelope encoding might also lead to a higher inter-individual response consistency, potentially explaining the higher correlations between the gerbils' response latencies post-exposure compared to pre-exposure (Fig 8). Improved coherence and temporal precision have also been reported for ANF responses in a mouse model of NIHL and were hypothesized to be caused by changes in synaptic transmission [78], potentially due to an increase in ribbon size after noise exposure [79].

Taken together, noise overexposure can lead to reduced peripheral input that mainly affects the high-frequency synaptopathic range, combined with a compensatory increase in central gain that causes enhanced envelope encoding and hypersensitivity particularly to low-frequency regions. These noise exposure effects might be favorable for vowel discrimination at certain stimulation levels, since the discrimination of vowels is mainly based on the distinct low-frequency formant patterns reflected in the temporal envelope of vowels (for both humans and gerbils) [31,80].

### Maintained consonant discriminability as a result of opposing noise effects

In contrast to the improvement in the gerbils' overall vowel discrimination ability from before noise overexposure to after noise overexposure, the average *d'*-values (Fig 4A) and response latencies (Figs 4B and 7B) quantifying the gerbils' overall consonant discrimination ability remained unaltered by the noise exposure. In the more detailed analysis of the response latencies for the different consonant types, we saw that as the only exceptions, noise exposure

selectively shortened the response latencies for discriminations between consonants with the same place of articulation as well as between consonants with the same voicing and the same place of articulation (Fig 10). In both cases, the manner of articulation of the consonants being discriminated would be different, raising the possibility that the discriminability of different manners of articulation was improved post-trauma compared to pre-trauma. Accordingly, the acoustic trauma selectively increased the discriminability of consonants with these specific combinations of articulatory features.

Additionally, we observed a permanent noise-induced reduction in ABR wave I amplitude in response to clicks (Fig 2A), which is probably due to the observed loss and pathological changes of synapses between IHCs and ANFs in the high-frequency region within and above the exposure noise band (Fig 3). As a consequence of the synaptopathy, there is probably a reduced input from these high-frequency regions. Following the idea of a compensatory overall increase in central gain, there might not only be an improvement in the gerbils' vowel discrimination ability due to enhancements in envelope and low-frequency encoding, but high-frequency encoding and consequently also the gerbils' consonant discrimination ability might be restored in a way that it matches pre-exposure levels. This corresponds to the observations from Monaghan et al. (2020) that the discrimination of speech sounds in the gerbil model for NIHL was most impaired for sounds with larger proportions of energy in high-frequency regions within and above the exposure noise band [51]. However, in contrast to our results, they found that beyond vowel discriminability, also overall consonant discriminability was improved post-trauma for a stimulation level of 60 dB SPL (in contrast to a decline at 75 dB SPL). They argued that the frequency-specific detrimental effects of the noise exposure were ameliorated by the overall increase in neural gain [51], since low-SR ANFs, whose synapses are thought to be the main target of the noise-induced synaptopathy [12], are of rather little relevance at moderate sound intensities. Given the fact that the stimulation level in the present study was higher (65 dB SPL), a loss of low-SR ANFs may have induced more adverse effects leading to an overall unchanged consonant discrimination ability.

Another factor that may have contributed to an overall unaltered consonant discrimination ability is the noise-induced enhancement in envelope encoding that was discussed previously. It was suggested that abnormally enhanced envelope encoding may be distracting for the discrimination of fine-scale speech sounds as consonants [71,75,77]. In this way, the strong representation of the temporal envelope, potentially ameliorating the encoding of the masking noise and the vowels, may have led to a decrease in relative consonant discrimination ability. In combination with the increase in central gain, this may ultimately result in an unchanged overall consonant discrimination ability.

### Effects of NIHL on speech discriminability might depend on stimulation level and configuration of noise exposure

In the present study, we observed that the overall speech sound discrimination ability of gerbils was either unchanged or even improved (depending on the type of speech sounds) after they were exposed to an acoustic trauma and suffered from NIHL with a TTS. This stands in contrast to the commonly suggested link between noise overexposure and speech-in-noise perception difficulties in humans. What might be the reasons for these differences?

As argued above, speech sound discriminability may not only depend on the SNR, but it has further been reported to be level-dependent with larger declines for higher stimulation intensities [51,81]. Thus, the tested speech sound level of 65 dB SPL and noise masker level of 60 dB SPL might have been too low for eliciting degradations of speech-in-noise perception as they are experienced by human listeners in noisy environments. Indeed, the typical scenario for speech-in-noise difficulties in human listeners potentially comprises higher overall sound intensities and lower or more variable SNRs with multiple competing sound sources and modulated background noises. It was hypothesized that listeners with sensorineural hearing loss might especially have problems with the segregation of speech from fluctuating backgrounds, since the enhanced envelope encoding resulting from noise overexposure might lead to an unfavorable enhancement of the modulated background noise [77]. The ICRA-1 noise that was used in the present study is speech-shaped but not amplitude-modulated, so that this, potentially negative, effect of the noise exposure did not come into effect in our study.

Additional aspects of the experimental design that might have led to differences in the noise effects compared to humans suffering from sensorineural hearing loss are the artificial nature of the noise exposure and the usage of anesthesia. NIHL in humans most often refers to a condition that occurs as a consequence of a long period of continuous or repeated exposure to loud noises, for example due to occupational noise exposure in factories or military service. Thus, even though NIHL can be observed following a rather short single exposure to a loud noise under the influence of anesthesia in an experimental environment, it is still a rather artificial condition in contrast to what happens to noise-exposed human listeners. For example, noise-induced damage may normally be distributed more evenly across a wider range of frequencies [82,83], since human listeners are likely exposed to a variety of loud sounds with different frequencies over the lifetime [51]. Consequently, NIHL as experienced by humans might be overall more severe and may differ in certain aspects from what can be mimicked in gerbils under experimental conditions. Also, NIHL in humans is often accompanied by age-related hearing loss in elderly subjects [84], so that declines in hearing abilities may also be the result of age-related deteriorations or mixed effects of noise- and age-related changes in the auditory system, whereas young-adult gerbils were used in the present study that should not be affected by any age-related declines in addition to the NIHL. Additionally, it has been shown that speech perception errors in human listeners occur mostly due to reduced abilities to use high-frequency cues for the discrimination of consonants rather than reduced abilities to use low-frequency cues for the discrimination of vowels [67]. Thus, potential declines in consonant discrimination ability might have more severe practical effects in speech-in-noise perception in humans than expected from experiments with gerbils.

## Conclusion

All in all, even though there are some differences between the experimentally provoked NIHL of gerbils and the real-world conditions of noise-exposed human listeners, the present study offered insights into the effects of a well-defined noise exposure on speech sound discrimination in a model organism with a very human-like perception of vowels and consonants [31]. In this regard, the results provide meaningful findings about noise exposure effects on speech-in-noise perception under controlled conditions that might also contribute to the conditions experienced by human listeners with NIHL.

The aim of the present study was to investigate the commonly suggested link between noise exposure, synaptopathy and speech-in-noise perception difficulties. The here applied acoustic trauma caused NIHL with a TTS and potentially pathological synaptic changes in the exposed gerbils. However, the behavioral consonant discrimination ability of the gerbils did not decline and the vowel discrimination ability even improved post-exposure. Thus, we cannot confirm the connection between noise exposure and speech-in-noise perception difficulties – at least not under all circumstances and for the gerbil model of NIHL. Nevertheless, there is evidence that the effect of noise exposure and NIHL on speech perception depends on a variety of different factors as the stimulation level, the masking noise and the exact acoustic trauma conditions. Consequently, there might be (larger and/or negative) noise exposure effects on speech-in-noise perception for higher stimulation levels, modulated masking noises or NIHL configurations with wider affected frequency regions. Especially consonant discriminability might be affected more severely under more adverse circumstances. Still, our results show that noise exposure and temporary NIHL are not necessarily linked to speech-in-noise perception difficulties and that compensatory central mechanisms as a result of the reduced peripheral input can even lead to an enhanced speech sound discriminability in some cases.

## Supporting information

**S1 Fig. Influence of acoustic trauma on hit rate and false alarm rate.** Hit rates (**A**) for the behavioral performance of the gerbils ($n = 9$) were significantly higher for CVC conditions (but not for VCV conditions) post-trauma compared to pre-trauma. Neither the noise trauma nor the logatome type had an effect on the false alarm rate (**B**). *: $p < 0.05$, ***: $p < 0.001$. (TIF)

## Acknowledgments

We thank Chieh-Ju Chi and Dogus Özgün Ulukuz for their contribution to collection of the gerbil data and their care for the animals during the data collection period.

## Author contributions

**Conceptualization:** Carolin Jüchter, Georg Klump.

**Data curation:** Carolin Jüchter, Rainer Beutelmann, Sonny Bovee, Katja Bleckmann.

**Formal analysis:** Carolin Jüchter.

**Funding acquisition:** Georg Klump.

**Investigation:** Carolin Jüchter.

**Methodology:** Carolin Jüchter, Rainer Beutelmann, Sonny Bovee, Katja Bleckmann, Georg Klump.

**Project administration:** Carolin Jüchter, Georg Klump.

**Resources:** Georg Klump.

**Software:** Rainer Beutelmann.

**Supervision:** Georg Klump.

**Validation:** Carolin Jüchter, Rainer Beutelmann, Georg Klump.

**Visualization:** Carolin Jüchter.

**Writing – original draft:** Carolin Jüchter.

**Writing – review & editing:** Carolin Jüchter, Rainer Beutelmann, Georg Klump.

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
