## [Decision Letter · Decision Letter 0]

11 Mar 2025

PONE-D-25-05066Speech-in-noise discriminability after noise exposure: Insights from a gerbil model of acoustic traumaPLOS ONE

Dear Dr. Klump,

Thank you for submitting your manuscript to PLOS ONE. After careful consideration, we feel that it has merit but does not fully meet PLOS ONE’s publication criteria as it currently stands. Therefore, we invite you to submit a revised version of the manuscript that addresses the points raised during the review process.

We look forward to receiving your revised manuscript.

Kind regards,

Andreas Buechner

Academic Editor

PLOS ONE

Journal Requirements:

“This work was funded by the Deutsche Forschungsgemeinschaft (DFG, German Research Foundation) under Germany's Excellence Strategy – EXC 2177/1 (Project-ID 390895286).”

3. We noted in your submission details that a portion of your manuscript may have been presented or published elsewhere. “The pre-trauma data was also used as part of a comparison of behavioral data from young-adult and quiet-aged gerbils [https://doi.org/10.1101/2024.11.06.622262] and small parts of the datasets, that is, data for behavioral discriminations between the vowels /aː/, /eː/ and /iː/ from five gerbils were used for a comparison with data from single ANF recordings in a recent study [https://doi.org/10.3389/fnins.2023.1238941].”

Reviewers' comments:

Reviewer's Responses to Questions

**Comments to the Author**

1. Is the manuscript technically sound, and do the data support the conclusions?

Reviewer #1: Partly

Reviewer #2: Partly

2. Has the statistical analysis been performed appropriately and rigorously? 

Reviewer #1: Yes

Reviewer #2: No

3. Have the authors made all data underlying the findings in their manuscript fully available?

Reviewer #1: Yes

Reviewer #2: Yes

4. Is the manuscript presented in an intelligible fashion and written in standard English?

Reviewer #1: Yes

Reviewer #2: Yes

5. Review Comments to the Author

Reviewer #1: Noise-induced hearing loss is thought to be the cause of speech perception difficulties, particularly in the presence of background noise. When such deficits in speech-in-noise perception are not reflected by an altered pure-tone audiometry, the condition is referred to as a hidden hearing loss. As animal models offer the possibility of a precise experimental approach with well-defined settings, this study investigates speech-in-noise discrimination in gerbils and how the discriminability of different types of vowels and consonants changes from before to after noise overexposure. Nine gerbils were trained to discriminate a deviant consonant-vowel-consonant (CVC) or vowel-consonant-vowel (VCV) combination in a sequence of CVC or VCV standards, respectively. The gerbils were better at discriminating vowels than consonants. While the ability to discriminate vowels was improved after noise exposure, along with a reduction in reaction time, the ability to discriminate consonants was rather poor before and after noise exposure. This is in contrast to the commonly suggested link between noise exposure and speech-in-noise perception difficulties in humans. Nevertheless, there is no real conclusion in the abstract.

For someone outside the field, this work is very specific and the amount of work may be better appreciated in a more specialised journal. However, as the topic is of great importance to the healthcare system, it might still fit well in a journal such as Plos one.

Here are some major concerns that need to be addressed:

- Although the authors aimed to investigate the commonly suggested link between noise overexposure and speech-in-noise perception difficulties, they used only one background noise, i.e., the masking sound “steady-state speech shaped noise”.

- Please superimpose the individual data points for each animal on the box plots.

- The authors praise the use of animals as a precise experimental approach with well-defined noise exposure to investigate the relationship between noise exposure and synaptopathology, but they do not verify synaptopathology by immunohistological staining.

- What was the time interval given to the gerbils for a correct response? Could it be that the response time for the VCV reflects the average of random poking (together with a rather poor D-prim of 1), whereas the shorter latencies reflect the precise response to the signal (together with a D-prim of 1.6)?

- Gerbils were exposed to a CVC in a sequence of VCV. What was the ratio (e.g. 20% CVC and 80% VCV)?

- It is very unsatisfactory to refer to another manuscript for the methods. I would appreciate a representative example of CVC in VCV or VCV in CVC. How long were the gerbils trained and to what level of performance, e.g. correctly answering the CVC while rejecting the VCV by 70%, which would result in a D-prime of about 1? Were the gerbils trained for a specific criterion or for a specific time? How long were the gerbils trained per day? What was the "inter-vowel interval"?

- 14 gerbils were exposed to noise trauma, but only 9 were reported for behavioural outcomes. What happened to the 5 excess gerbils?

- No 'naive' control or sham-exposed group was used. Since all the gerbils were exposed to noise-induced hearing loss and the ABR showed only a transient elevated threshold, could it be that the behaviour of the gerbils in terms of CVC simply improved over time, while the VCV remained just above chance level before and after noise exposure?

- Please also show some kind of raw data, but not just the D prime. In addition, catch trials in which the target logatome was the same as the reference logatome were used as a measure of spontaneous responding. Did the authors report these sequences? Was this parameter included anywhere in the calculation?

- I just don't understand the "perception maps", which may - at least in part - be related to the poor quality/resolution of the images. I find the confusion matrices very confusing! Why is 93% of the explained variance in the MDS a "goodness of fit"?

- I do not understand why four CVC conditions were used to test the discriminability of vowels, whereas three VCV conditions were used to test the discriminability of consonants. Also, two-dimensional perceptual maps were used for the vowels, but three-dimensional perceptual maps were used for the consonants. Why?

- What was the cage size? What were the reward pellets made of? How much regular food were the gerbils given per day? Was it possible to feed them in a group?

Reviewer #2: The study by Jüchter et al. compares behavioral speech (vowel and consonant) discrimination data from Mongolian gerbils with and without noise overexposure. The noise exposure is presumed to cause synaptopathy of spiral ganglion neurons based on electrophysiology (ABR). Following exposure, behavioral performance (discriminability and response latency) was not reduced; rather, it improved for vowel stimuli.

While the behavioral data are impressive, I have several major concerns about the paradigm and interpretation, as detailed below.

Major Concerns:

> Extreme Noise Exposure: The noise exposure is quite extreme (115 dB SPL). Generally, this level of overexposure causes moderate hearing loss in several species, including gerbils. The authors discuss this, but not in a satisfactory way. What was the weighting of the sound level (A, B, or C-weighted)? Was the sound level measured at the speaker, and could it have dropped near the ear?

> Limited Hearing Loss in Relevant Frequency Range: There is little hearing loss (including temporary threshold shift) below 4 kHz, a region largely sufficient for robust vowel and consonant perception. For example, the first two formants (generally < 2 kHz) convey substantial information about vowels. While the authors appropriately discuss a number of mechanisms, in my view, these data seem to be largely consistent with gerbils getting overtrained without any hearing loss in the frequency range of interest. In any case, the fact that the temporary threshold shift was only limited above 4 kHz and not at lower frequencies should be mentioned in the abstract.

> Response Latency vs. d-prime/Percent Correct: The study, consistent with previous work by the authors, uses response latency as a key metric. However, some details/rationale are missing. How was the response latency calculated? For example, for /CVC/, was it measured from /C/ onset or /V/ onset? If from the onset of the whole logatome, it would make sense that /VCV/ response latency is longer than /CVC/ because vowels are longer than consonants. For MDS, was the latency limited only to correct trials? Additionally, to my knowledge, response latency tracks task difficulty and is influenced by cognitive effort (which is interesting). However, a more direct metric would be to use the confusion matrix (percent correct or d-prime) for each pair.

> Sound Level of Logatomes: Consonants like fricatives are up to 40 dB softer than vowels (Stevens 2000). What do 60 dB for speech-shaped noise and 65 dB SPL for logatomes mean? Is it RMS-normalized for the whole segment? What about for /VCV/? Assuming the whole token is at 65 dB SPL, which is presumably dominated by the vowel, fricatives could be 35-45 dB SPL (which means -25 to -15 dB SNR). This task would be extremely difficult, if not impossible.

Additional Concerns:

The overall structure of the manuscript is good, but the writing is vague/missing details at several places. For example:

> Abstract: "However, the connection between noise overexposure and deteriorated speech-in-noise perception is not clear yet." We know various forms of SNHL cause speech-in-noise perception (e.g., Festen and Plomp 1990). The statement should be made specific to hidden hearing loss.

> Figure 6: What is each dot? Assuming each vowel/consonant pair (pooled across all animals), but it would be good to spell this out. In general, figure captions would benefit from more description about what is plotted and what the error bars represent.

> First Paragraph of Conclusion (L602-610): Quite redundant as presented; generally, this section should be a summary of key findings.

> L827-835: A significant portion of the conclusion is about alternatives not related to the present study.

> Use of "Hearing Loss": Several places in the manuscript mention there was noise-induced hearing loss. Hearing loss generally refers to shifts in the audiogram. Here, ABR thresholds (proxies for audiograms) are not different following noise exposure.

6. PLOS authors have the option to publish the peer review history of their article (what does this mean? ). If published, this will include your full peer review and any attached files.

**Do you want your identity to be public for this peer review?** For information about this choice, including consent withdrawal, please see our Privacy Policy .

Reviewer #1: No

Reviewer #2: No

---

## [Author Response · Author response to Decision Letter 1]

1 Jul 2025

Reviewer #1:

Noise-induced hearing loss is thought to be the cause of speech perception difficulties, particularly in the presence of background noise. When such deficits in speech-in-noise perception are not reflected by an altered pure-tone audiometry, the condition is referred to as a hidden hearing loss. As animal models offer the possibility of a precise experimental approach with well-defined settings, this study investigates speech-in-noise discrimination in gerbils and how the discriminability of different types of vowels and consonants changes from before to after noise overexposure. Nine gerbils were trained to discriminate a deviant consonant-vowel-consonant (CVC) or vowel-consonant-vowel (VCV) combination in a sequence of CVC or VCV standards, respectively. The gerbils were better at discriminating vowels than consonants. While the ability to discriminate vowels was improved after noise exposure, along with a reduction in reaction time, the ability to discriminate consonants was rather poor before and after noise exposure. This is in contrast to the commonly suggested link between noise exposure and speech-in-noise perception difficulties in humans. Nevertheless, there is no real conclusion in the abstract.

We added a conclusive sentence to the abstract (l. 46ff). Further changes have been made to the abstract in order to meet the word limit.

For someone outside the field, this work is very specific and the amount of work may be better appreciated in a more specialised journal. However, as the topic is of great importance to the healthcare system, it might still fit well in a journal such as Plos one.

Here are some major concerns that need to be addressed:

- Although the authors aimed to investigate the commonly suggested link between noise overexposure and speech-in-noise perception difficulties, they used only one background noise, i.e., the masking sound “steady-state speech shaped noise”.

It is correct that only one background noise was used, however, we do not see why this constitutes a major concern. As described in the methods part (l. 253ff), the background noise (ICRA-1) is a continuous steady-state noise masker with a speech-shaped spectrum, which is an established noise masker that is often used in studies investigating speech perception. Using ICRA-1 as a masker further enables a direct comparison with previous studies that also used OLLO logatomes in combination with the ICRA-1 masker (e.g., Jürgens & Brand, J Acoust Soc Am. 2009; doi: 10.1121/1.3224721; Meyer et al., J Acoust Soc Am. 2010; doi: 10.1121/1.3493450).

- Please superimpose the individual data points for each animal on the box plots.

The data points for the individual animals were superimposed on the boxplots in the revised figures.

- The authors praise the use of animals as a precise experimental approach with well-defined noise exposure to investigate the relationship between noise exposure and synaptopathology, but they do not verify synaptopathology by immunohistological staining.

We now performed immunohistochemical investigations of the synapses and added our findings regarding synaptopathy to the manuscript. We adjusted different parts of the methods (l. 256ff), results (l. 392ff) and discussion (l. 735ff & l. 836ff) accordingly.

- What was the time interval given to the gerbils for a correct response? Could it be that the response time for the VCV reflects the average of random poking (together with a rather poor D-prim of 1), whereas the shorter latencies reflect the precise response to the signal (together with a D-prim of 1.6)?

The gerbils had to indicate a detected target logatome by jumping off the pedestal within 1.5 seconds from the onset of the target stimulus. This information was now added to the manuscript (l. 221ff)

The longer response latencies (and lower d’-values) for the VCV conditions compared to CVC conditions cannot be explained through a larger number of random responses by the gerbils, since the false alarm rate was similar for CVC and VCV conditions, but they directly reflect differences in hit rates (not shown in the manuscript, now added as supplementary material S1 Fig).

- Gerbils were exposed to a CVC in a sequence of VCV. What was the ratio (e.g. 20% CVC and 80% VCV)?

As mentioned in l. 240ff, only a change in the middle phoneme of the logatomes had to be detected so that the discriminability was always tested between logatomes with the same phonetic context. Thus, gerbils were never exposed to a CVC in a sequence of VCVs. We added an example sequence for clarification to the text (l. 242ff).

- It is very unsatisfactory to refer to another manuscript for the methods. I would appreciate a representative example of CVC in VCV or VCV in CVC. How long were the gerbils trained and to what level of performance, e.g. correctly answering the CVC while rejecting the VCV by 70%, which would result in a D-prime of about 1? Were the gerbils trained for a specific criterion or for a specific time? How long were the gerbils trained per day? What was the "inter-vowel interval"?

We understand that it is more convenient if the methods part does not refer to another paper, however, given the high complexity of the behavioral paradigm that is described in detail in our previous paper, we decided to limit the methodological information in this paper to the essentials, since the length of the current paper would otherwise increase considerably. We still added some more important information about the experimental setup (l. 211ff), behavioral paradigm (l. 215ff) and data analysis (l. 307ff), which can now be found directly in the present manuscript.

As mentioned in the previous comment, CVCs and VCVs were never mixed in one experimental condition.

The training procedure took between 3 and 8 weeks for the different gerbils, starting with the discrimination of simple pure tones, slowly increasing the complexity of the stimuli that needed to be discriminated (now mentioned in l. 223ff). Data collection started when the gerbils reached a stable performance level with the logatomes (stable hit rates and false alarm rates) with the validity criterion that average d’-values of complete sessions had to be larger than 0.5.

Single experimental sessions typically lasted between 20 and 60 minutes and each gerbil was tested about 1 to 3 times a day for 5 days a week (now mentioned in l. 230f & l. 120f)

Logatomes were presented every 1.3 seconds (now mentioned in l. 216ff).

- 14 gerbils were exposed to noise trauma, but only 9 were reported for behavioural outcomes. What happened to the 5 excess gerbils?

The five “excess” gerbils were part of a different cohort of gerbils that was only used to evaluate the effects of different acoustic trauma conditions on the ABR, but which were never trained for any behavioral experiments. The “main” cohort of nine gerbils was trained for the behavioral experiments and eventually underwent the same acoustic trauma and ABR measurement scheme as the other five gerbils. Finally, we pooled the ABR data from both gerbil groups for the current study in order to increase the animal number (and thus statistical power) for the investigation of the ABR data. There are no gerbils that were trained for the behavioral experiments and whose data was not analyzed or used for the manuscript. This was now clarified in the manuscript (l. 107ff).

- No 'naive' control or sham-exposed group was used. Since all the gerbils were exposed to noise-induced hearing loss and the ABR showed only a transient elevated threshold, could it be that the behaviour of the gerbils in terms of CVC simply improved over time, while the VCV remained just above chance level before and after noise exposure?

No ’naive’ or sham-exposed control group was used, since the gerbils were used as their own controls (pre-trauma vs. post-trauma). The gerbils’ performances for the discrimination of both CVCs and VCVs (with d’-values around 1.0 to 1.5) were far better than chance level (which would correspond to a d’ = 0). Thus, we are convinced that the gerbils were working under stimulus control at all times and that differences in d’-values reflect actual changes in discrimination abilities.

In the case of simple improvements over time or persistent training effects we would rather expect improvements for discriminations of all types of logatomes over time and no selective effect for a specific logatome type.

- Please also show some kind of raw data, but not just the D prime. In addition, catch trials in which the target logatome was the same as the reference logatome were used as a measure of spontaneous responding. Did the authors report these sequences? Was this parameter included anywhere in the calculation?

We show the response latencies, which is the raw data. Further raw data are the hit rates and false alarm rates, however, since the d’ is based on these two measures (as described in l. 304ff), we decided to limit the data shown in the main manuscript to the response latencies and d’-values to avoid redundancy. Hit rates and false-alarm rates were now added to the supplementary material (S1 Fig). Further, the raw data was now made freely accessible via zenodo (https://doi.org/10.5281/zenodo.15489868).

- I just don't understand the "perception maps", which may - at least in part - be related to the poor quality/resolution of the images. I find the confusion matrices very confusing! Why is 93% of the explained variance in the MDS a "goodness of fit"?

The pdf version of the manuscript that is provided by PLOS ONE only contains low-resolution versions of the figures, but there is a link in the upper right corner on each page with a figure that provides access to the high-resolution version of the figure.

Confusion matrices served as a basis for the further analysis of the response latencies. The matrices were filled with the response latencies for all combinations of reference and target logatomes of the subjects. In this way, the discrimination performance was quantified, with short response latencies indicating salient differences between reference and target logatomes, whereas long response latencies suggest a minor discriminability. (These information and further details can be found in the paper that we refer to in the methods part)

A goodness of fit measure determines whether a model is a good fit to the underlying data. In our case, the perceptual map generated by the MDS was able to explain 93% of the variance in the underlying response latencies, which means that the perceptual map represented and visualized the measured differences in response latencies between the different logatomes very well. Thus, we used the amount of explained variance to quantify how well the perceptual maps reflected the differences in response latencies that we measured.

- I do not understand why four CVC conditions were used to test the discriminability of vowels, whereas three VCV conditions were used to test the discriminability of consonants. Also, two-dimensional perceptual maps were used for the vowels, but three-dimensional perceptual maps were used for the consonants. Why?

There was no specific reason for the number of CVC conditions vs. VCV conditions, except for similar total numbers of CVCs and VCVs: In each CVC condition, the discriminability between 10 different vowels was tested, while in each VCV condition, the discriminability between 12 different consonants was tested. As we wanted to test the discriminability of both vowels and consonants in the context of different flanking consonants/vowels (i.e., different conditions), the choice of 4 CVC conditions and 3 VCV conditions resulted in a similar number of total CVCs and VCVs tested (40 CVCs vs. 36 VCVs, respectively).

As written in l. 299ff, three-dimensional perceptual maps were used for the consonants in contrast to two-dimensional maps for the vowels in order to reach similar goodness-of-fit values for the perceptual maps of vowels and consonants, with more than 93% of explained variance in the MDS solutions. This ‘need’ for an additional dimension can be explained by the more complex structure of consonants in comparison to vowels, which can be represented more accurately in three dimensions.

- What was the cage size? What were the reward pellets made of? How much regular food were the gerbils given per day? Was it possible to feed them in a group?

The gerbils were held in EU type IV cages (590 x 380 x 200 mm; 1,815 cm2 floor size). This is now mentioned in the manuscript (l. 114ff).

The reward pellets are made of spelt flour, ground rodent dry found, sunflower oil and water.

The amount of food given per day varied (largely) for the different gerbils and was adjusted every day depending on multiple factors such as their body weight and the amount of reward pellets that they got during the experimental sessions. The average amount of food a gerbil was given per day ranged approximately from 2 to 6 g. As mentioned in l. 121f, the gerbils’ body weights were kept at about 90% of their free feeding weights.

Gerbils were usually fed together with their cage mates. Only in rare cases (e.g., unexpected weight loss or illness) gerbils were fed separately.  

Reviewer #2:

The study by Jüchter et al. compares behavioral speech (vowel and consonant) discrimination data from Mongolian gerbils with and without noise overexposure. The noise exposure is presumed to cause synaptopathy of spiral ganglion neurons based on electrophysiology (ABR). Following exposure, behavioral performance (discriminability and response latency) was not reduced; rather, it improved for vowel stimuli. While the behavioral data are impressive, I have several major concerns about the paradigm and interpretation, as detailed below.

Major Concerns:

> Extreme Noise Exposure: The noise exposure is quite extreme (115 dB SPL). Generally, this level of overexposure causes moderate hearing loss in several species, including gerbils. The authors discuss this, but not in a satisfactory way. What was the weighting of the sound level (A, B, or C-weighted)? Was the sound level measured at the speaker, and could it have dropped near the ear?

A variety of methodological differences (exposure level, frequency, bandwidth, duration, experimental setup, anesthesia, age at traumatization) between different studies investigating acoustic traumata that may influence the severity of the noise overexposure effects are outlined in the discussion of the manuscript (l. 703ff). As mentioned there, we believe that particularly the usage of anesthesia and octave-band noise for traumatization may have led to relatively small threshold shifts in comparison to previous studies (l. 713ff). In other studies where more severe hearing loss was reported for similar exposure levels, awake animals were traumatized, meaning that protective effects of anesthetics were absent and susceptibility may have been higher due to additional stress responses in the noisy environment.

The sound level was calibrated with the microphone positioned at the position of the gerbil’s head during the traumatization. The geometry of the traumatization box precluded the occurrence of standing waves. Thus, the level will not have dropped near the ear. The sound level meter was set to C-weighting which includes the frequency range of the traumatization noise. The C-weighting has a negligible effect (i.e., below 1 dB) on the accuracy of the sound level reading the frequencies used for the noise trauma, 2-4 kHz.

> Limited Hearing Loss in Relevant Frequency Range: There is little hearing loss (including temporary threshold shift) below 4 kHz, a region largely sufficient for robust vowel and consonant perception. For example, the first two formants (generally < 2 kHz) convey substantial information about vowels. While the authors appropriately discuss a number of mechanisms, in my view, these data seem to be largely consistent with gerbils getting overtrained without any hearing loss in the frequency range of interest. In any case, the fact that the temporary threshold shift was only limited above 4 kHz and not at lower frequencies should be mentioned in the abstract.

We added to the abstract that there was a temporary threshol

---

## [Decision Letter · Decision Letter 1]

5 Aug 2025

Speech-in-noise discriminability after noise exposure: Insights from a gerbil model of acoustic trauma

PONE-D-25-05066R1

Dear Dr. Klump,

We’re pleased to inform you that your manuscript has been judged scientifically suitable for publication and will be formally accepted for publication once it meets all outstanding technical requirements.

Kind regards,

Andreas Buechner

Academic Editor

PLOS ONE

Additional Editor Comments (optional):

Reviewers' comments:

Reviewer's Responses to Questions

**Comments to the Author**

1. If the authors have adequately addressed your comments raised in a previous round of review and you feel that this manuscript is now acceptable for publication, you may indicate that here to bypass the “Comments to the Author” section, enter your conflict of interest statement in the “Confidential to Editor” section, and submit your "Accept" recommendation.

Reviewer #1: All comments have been addressed

Reviewer #2: All comments have been addressed

2. Is the manuscript technically sound, and do the data support the conclusions?

Reviewer #1: Yes

Reviewer #2: Yes

3. Has the statistical analysis been performed appropriately and rigorously? 

Reviewer #1: Yes

Reviewer #2: Yes

4. Have the authors made all data underlying the findings in their manuscript fully available?

Reviewer #1: Yes

Reviewer #2: (No Response)

5. Is the manuscript presented in an intelligible fashion and written in standard English?

Reviewer #1: Yes

Reviewer #2: Yes

6. Review Comments to the Author

Reviewer #1: (No Response)

Reviewer #2: The authors have addressed all my concerns. Addition of synapse count strengthens the manuscript. Other than one wording suggestion (see below), I don't have any other concern. I thank the authors for their contribution to the field.

Abstract last sentence: Altogether, the results show -> Altogether, the results suggest

7. PLOS authors have the option to publish the peer review history of their article (what does this mean? ). If published, this will include your full peer review and any attached files.

**Do you want your identity to be public for this peer review?** For information about this choice, including consent withdrawal, please see our Privacy Policy .

Reviewer #1: No

Reviewer #2: No

---

## [Editor Report · Acceptance letter]

PONE-D-25-05066R1

PLOS ONE

Dear Dr. Klump,

I'm pleased to inform you that your manuscript has been deemed suitable for publication in PLOS ONE. Congratulations! Your manuscript is now being handed over to our production team.

Kind regards,

on behalf of

Andreas Buechner

Academic Editor

PLOS ONE